# Tissue and cellular rigidity and mechanosensitive signaling activation in Alexander disease

Liqun Wang[1], Jing Xia[2], Jonathan Li[3], Tracy L. Hagemann[4], Jeffrey R. Jones[4], Ernest Fraenkel [3], David A. Weitz[2,5], Su-Chun Zhang[4,6], Albee Messing[4,7] & Mel B. Feany[1]

Glial cells have increasingly been implicated as active participants in the pathogenesis of neurological diseases, but critical pathways and mechanisms controlling glial function and secondary non-cell autonomous neuronal injury remain incompletely defined. Here we use models of Alexander disease, a severe brain disorder caused by gain-of-function mutations in GFAP, to demonstrate that misregulation of GFAP leads to activation of a mechanosensitive signaling cascade characterized by activation of the Hippo pathway and consequent increased expression of A-type lamin. Importantly, we use genetics to verify a functional role for dysregulated mechanotransduction signaling in promoting behavioral abnormalities and non-cell autonomous neurodegeneration. Further, we take cell biological and biophysical approaches to suggest that brain tissue stiffness is increased in Alexander disease. Our findings implicate altered mechanotransduction signaling as a key pathological cascade driving neuronal dysfunction and neurodegeneration in Alexander disease, and possibly also in other brain disorders characterized by gliosis.

[1] Department of Pathology, Brigham and Women's Hospital, Harvard Medical School, Boston, MA 02115, USA. [2] School of Engineering and Applied Science, Harvard University, Cambridge, MA 02138, USA. [3] Department of Biological Engineering, Massachusetts Institute of Technology, Cambridge, MA 02139, USA. [4] Waisman Center, University of Wisconsin-Madison, Madison, WI 53705, USA. [5] Department of Physics, Harvard University, Cambridge, MA 02138, USA. [6] Department of Neuroscience and Department of Neurology, School of Medicine and Public Health, University of Wisconsin-Madison, Madison, WI 53705, USA. [7] Department of Comparative Biosciences, School of Veterinary Medicine, University of Wisconsin-Madison, Madison, WI 53705, USA. Correspondence and requests for materials should be addressed to M.B.F. (email: mel_feany@hms.harvard.edu)

Mechanotransduction, the detection by cells and tissues of intracellular and extracellular mechanical stimuli and conversion of those signals into biochemical readouts, is crucial in multiple aspects of cellular function and tissue development. Homeostasis of mechanotransduction is therefore tightly controlled and involves coordinated regulation of extracellular matrix proteins, transmembrane integrin and focal adhesion molecules, intracellular cytoskeletal and signaling proteins, and nuclear responders[1]. Perturbations in mechanotransduction homeostasis can lead to significant cellular and tissue dysfunction. In the cardiovascular system, shear stress plays an important role in the development of large vessel atherosclerosis[2], and pathological cardiac hypertrophy in response to prolonged mechanical stress or genetic defects in genes encoding sarcoplasmic structural proteins eventually results in heart failure[3].

In the nervous system, elegant studies have demonstrated a critical role for matrix stiffness in controlling axonal outgrowth during development[4]. In the adult brain, increasing tissue stiffness correlates with the degree of anaplasia in glial tumors, and downstream signaling pathways that may control invasion, dedifferentiation, and hypoxia response in response to increased tissue stiffness have been identified[5]. Similarly, acute, and possibly chronic, disruption of mechanotransduction pathways occurs in the context of traumatic brain injury[6]. However, the functional role that altered tissue stiffness plays in brain disorders is not well understood.

Here we use Alexander disease, a rare but severe neurological disorder caused by primary astrocyte dysfunction[7], to address the role of mechanotransduction signaling in the function and survival of brain cells. Alexander disease is caused by dominant missense mutations in the gene encoding a major intermediate filament protein of astrocytes, glial fibrillary acidic protein (GFAP)[8]. Alexander disease mutations appear to act via a gain-of-function mechanism because mice lacking GFAP are viable and have little distinctive neuropathology[9, 10]. In contrast, knock-in mice expressing disease-linked mutant human GFAP[11] recapitulate features of Alexander disease. In particularly these mice show robust aggregation of GFAP into the cytoplasmic eosinophilic, beaded Rosenthal fiber inclusions characteristic of the human disease. Rosenthal fibers are also commonly seen in the context of longstanding, dense reactive gliosis in human brains. Similarly, significant overexpression of human wild-type GFAP in mice causes Rosenthal fiber formation, and models other aspects of Alexander disease as well[11–16], raising the possibility that persistently elevated levels of wild-type human GFAP may cause cellular toxicity in some human brain disorders. However, increased levels of GFAP can also be observed in astrocytes that subserve protective functions, indicating a complex cellular interplay of neuroprotective and neurotoxic roles of astrocytes in normal and disease states[17–20].

To explore the role of mechanosensitive signaling pathways in astrogliopathy we combine here the strengths of analysis of

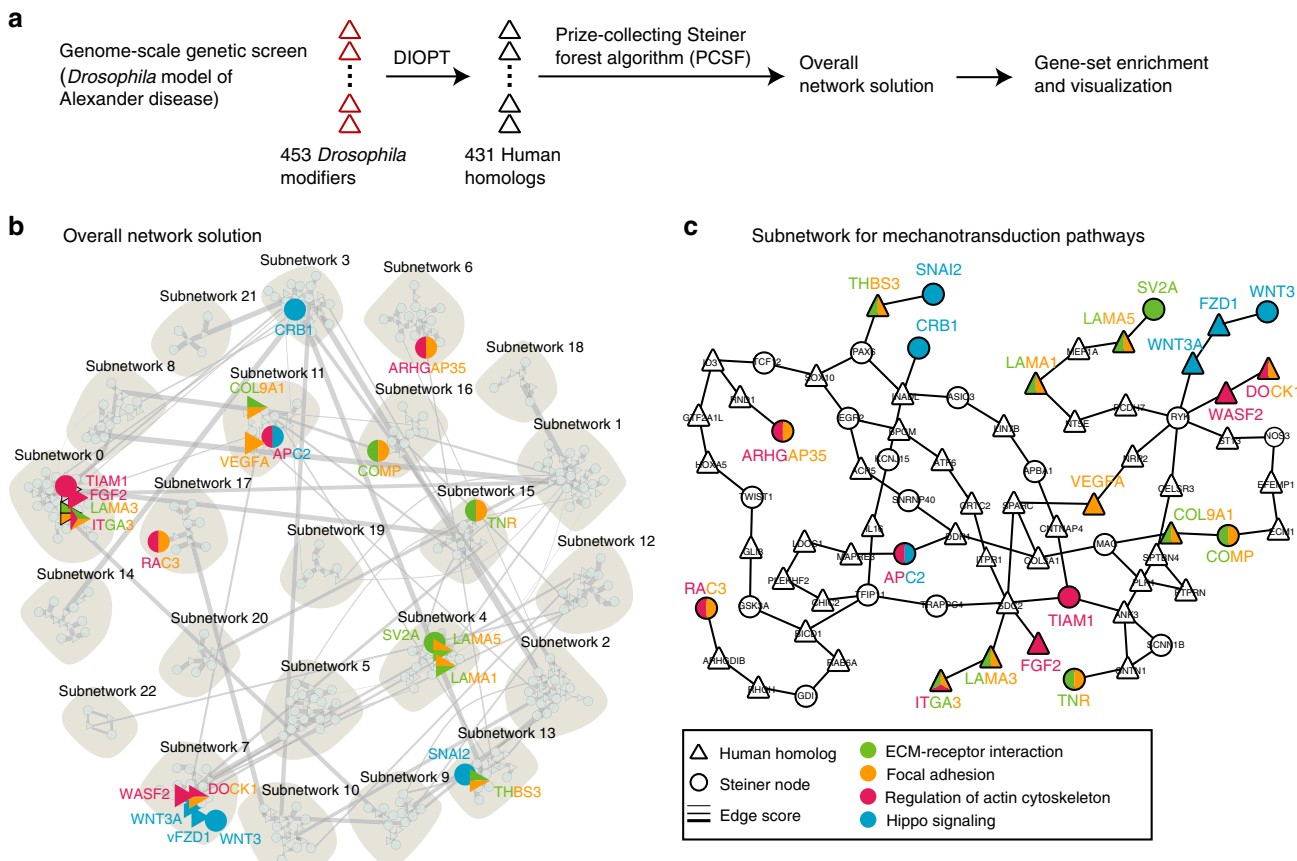

**Fig. 1** Genetic screen implicates mechanotransduction pathways. **a** Schematic representation of the data analysis for the results of a genome-scale genetic screen in the *Drosophila* model of Alexander disease. DIOPT: DRSC Integrative Ortholog Prediction Tool. **b** The overall network solution using the prize-collecting Steiner forest algorithm (PCSF) identifies pathways (green-, orange-, red-, and blue-filled nodes) implicated in mechanotransduction. Human homologs are represented as triangles; whereas, circles indicate Steiner nodes, or proteins that were not identified as modifiers in the fly screen, but are strongly implicated by the data and the PCSF. Edges indicate physical protein–protein interactions, with thickness being the confidence of the interaction based on experimental evidence compiled in iRefIndex13. **c** A subnetwork highlighting interactions of proteins implicated in mechanotransduction pathways

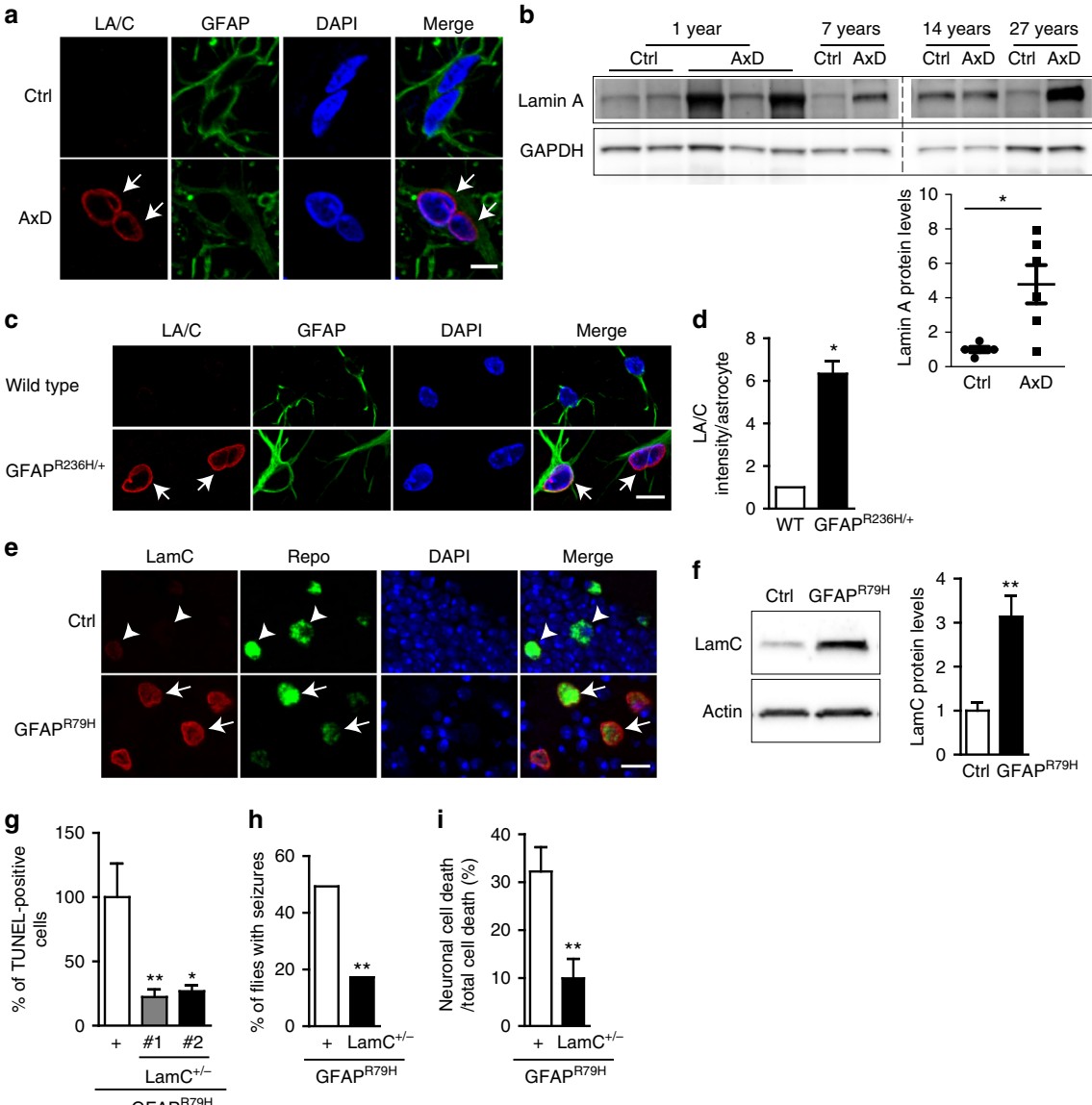

**Fig. 2** Increased expression of A-type laminin Alexander disease. **a** Double label immunofluorescence shows increased A-type lamin, lamin A/C (LA/C) expression in the astrocytes of a 1-year-old Alexander disease patient (AxD, arrows) compared to an age-matched control (Ctrl). GFAP labels astrocytes. Scale bar is 5 microns. **b** Western blots confirm a significant increase of lamin A expression in the white matter of Alexander disease patients compared to age-matched controls. The blots are reprobed for GAPDH to illustrate equivalent protein loading. $p = 0.0476$, Mann–Whitney test. **c** Double label immunofluorescence shows increased lamin A/C (LA/C) expression in the astrocytes of 3-month-old Alexander disease model mice ($GFAP^{R236H/+}$, arrows) compared to age-matched wild-type littermate control (wild type). GFAP labels astrocytes. Scale bar is 10 microns. **d** Quantification of immunofluorescence intensity shows marked increase of lamin A/C expression in 3-month-old Alexander disease model mice ($GFAP^{R236H/+}$) compared to age-matched wild-type mice (WT). $N = 3$ animals per genotype. A total of 30 astrocytes per animal were used for quantification. $p = 0.0126$, Wilcoxon test. **e** Double label immunofluorescence demonstrates increased expression of LamC, the *Drosophila* homolog of A-type lamin, in the glial cells of Alexander disease model flies ($GFAP^{R79H}$, arrows) compared to age-matched control flies (Ctrl, arrowheads). Repo marks glial cells. Scale bar is 5 microns. **f** Western blot demonstrates increased expression of LamC in Alexander disease model flies compared to age-matched controls. The blot is reprobed for actin to illustrate equivalent protein loading. $N = 8$, $p = 0.0002$, Mann–Whitney test. **g** Reducing LamC expression using two loss-of-function alleles of *LamC* in Alexander disease model flies markedly reduced cell death measured by TUNEL analysis. $N = 6$ per genotype. $*p < 0.05$, $**p < 0.01$, Kruskal–Wallis test. **h** Reducing LamC expression using a loss-of-function allele of *LamC* in Alexander disease model flies significantly decreased the number of flies with seizures. $N > 100$ per genotype. $p < 0.001$. $\chi^2$-test. **i** Reducing LamC expression using a loss-of-function allele of *LamC* in Alexander disease model flies rescues non-cell autonomous neuronal cell death. $N = 6$ per genotype. $p = 0.0152$, Mann–Whitney test. Flies are 20 days old in **e–g**, **i** and 1 day old in **h**

postmortem human brain tissue and experimental models of Alexander disease, including an Alexander disease-linked mutant GFAP knock-in mouse model[11], a *Drosophila* model of Alexander disease[21] and patient-specific astrocytes. We identify pervasive misregulation of mechanotransduction signaling pathways in Alexander disease, and show that genetic correction of these signaling rescues function and viability of neurons in vivo. Further, we demonstrate directly that brain stiffness is increased in the mouse model of Alexander disease. Taken together, these findings provide strong support for a causative role for perturbed glial mechanotransduction in promoting brain dysfunction in astrogliopathy.

## Results

**Genetic screen implicates mechanotransduction.** We have previously described a *Drosophila* model of Alexander disease based on expression of disease-linked forms of mutant human GFAP in fly glia. Our model recapitulates key features of the human disorder, including GFAP aggregation and non-cell autonomous neurodegeneration[21]. To gain novel insights into the pathogenesis of Alexander disease, we performed a genome-scale forward genetic screen in our *Drosophila* model. We reduced expression of 5375 genes, most (94%) conserved between flies and humans, in Alexander disease model animals and assessed toxicity using a sensitive and specific transgenic caspase reporter[22]. We identified 248 suppressors and 205 specific

enhancers of GFAP[R79H] toxicity (Supplementary Data 1). To investigate the cellular pathways identified by our modifiers, we used the solution of the prize-collecting Steiner forest algorithm (PCSF)[23] to map genetic modifiers onto a network of physical protein interactions using human interactome data (Fig. 1a). Interestingly, we found multiples nodes involved in mechanotransduction pathways, including extracellular matrix (ECM)-receptor interaction, focal adhesion, Hippo pathway, and actin cytoskeleton regulation (Fig. 1b, Supplementary Fig. 1 and Supplementary Table 1). A subnetwork generated by connecting pathway nodes illustrates the interactions among these pathways (Fig. 1c).

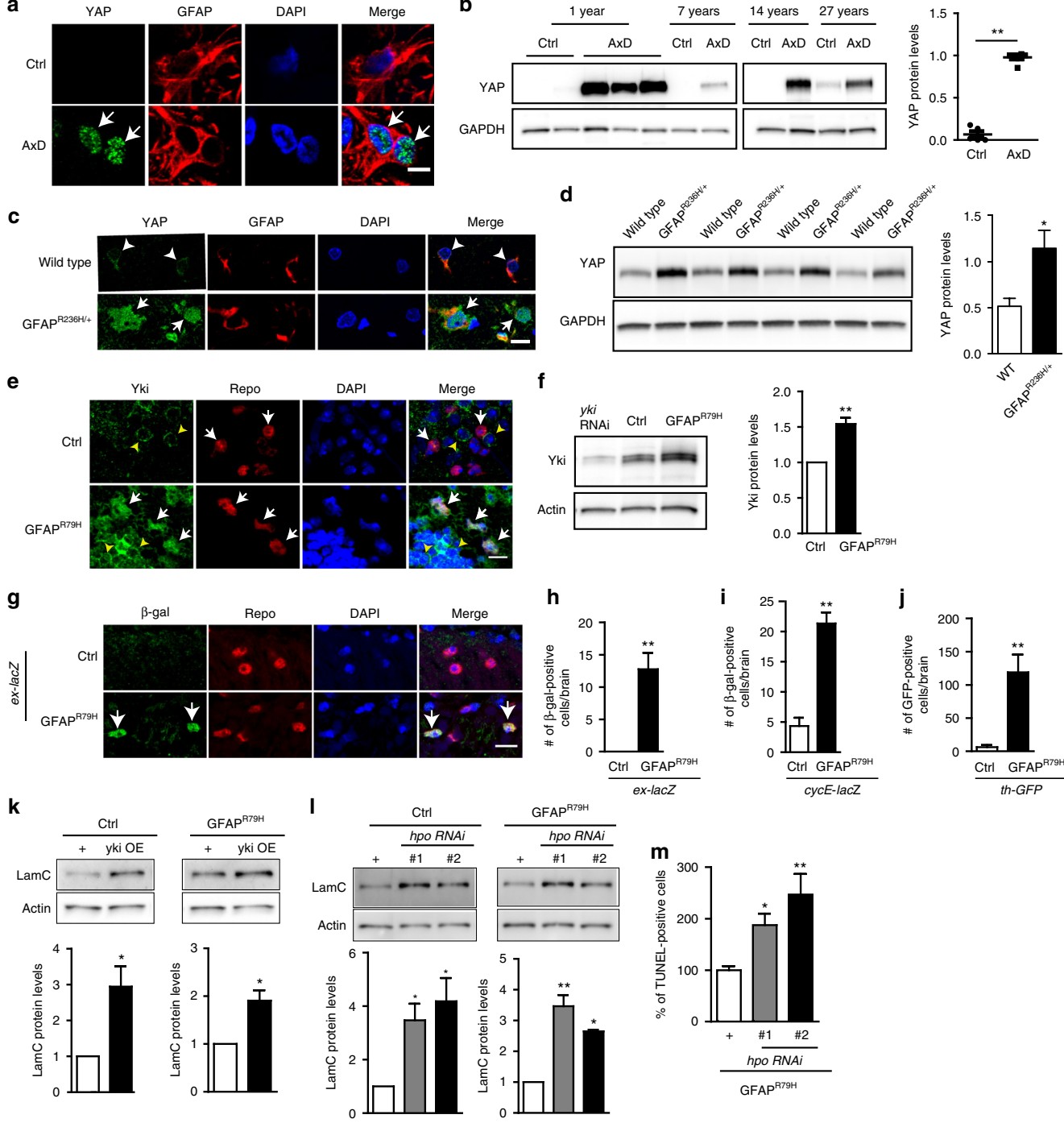

**Upregulation of lamin and YAP**. Recovery of multiple pathways involved in sensing and transducing mechanical signals in our genetic screen raised the possibility of a perturbation in mechanotransduction homeostasis in Alexander disease. To investigate this possibility in more detail, we first examined lamin proteins, which scale in expression with tissue stiffness and play a key role in mechanotransduction[1, 24]. Using double label immunofluorescence, we found strong lamin A expression in the astrocytes of Alexander disease patients, but very little lamin A in age-matched controls (Fig. 2a). Western blot analysis confirmed increased expression of lamin A in Alexander disease patient brains compared to age-matched control brains (Fig. 2b and Supplementary Fig. 7). Consistent with patient tissue data, we observed similar changes in a knock-in mouse model of Alexander disease[11]. Lamin A expression was significantly increased in the astrocytes of 3-month-old Alexander disease model mice compared to age-matched control mice (Fig. 2c, d). In contrast, we found no clear differences in lamin A in oligodendrocytes or neurons (Supplementary Fig. 2a-b, d-e). There was no detectable expression of lamin A in microglial cells as marked by Iba1 (Supplementary Fig. 2c,f). Lamin proteins are well conserved from *Drosophila* to mammals. Lamin C (LamC) is the sole *Drosophila* homolog of A-type lamin[25]. We found a significant increase in LamC expression in the glial cells of 20-day-old Alexander disease model flies compared to age-matched control flies (Fig. 2e, f and Supplementary Fig. 7), but not in 1-day-old flies (Supplementary Fig. 2g).

We next asked whether elevated LamC expression plays a causative role in GFAP toxicity using genetics. Analyzing cell death using terminal deoxynucleotidyl transferase (TdT) dUTP Nick-End Labeling (TUNEL) in Alexander disease model flies, we found that reducing LamC expression using loss-of-function alleles of *LamC* markedly reduced the number of TUNEL-positive cells (Fig. 2g). Western blots confirmed reduced LamC protein levels in these *LamC* mutants, with no change in GFAP[R79H] levels (Supplementary Fig. 2h,i). In addition, reducing LamC expression also significantly reduced the percentage of flies with seizures in Alexander disease model flies (Fig. 2h), confirming an important biological role of LamC expression in GFAP[R79H] toxicity. Non-cell autonomous neuronal loss is a prevalent feature of Alexander disease, which is recapitulated in our fly model[7, 21]. Using fluorescent double label for TUNEL and the neuronal marker Elav, we found that reducing LamC expression reduced both glial and neuronal cells death, with more pronounced rescue of neuronal loss (Fig. 2i and Supplementary Fig. 2j), suggesting that increased expression of LamC triggers neurodegeneration. To test this hypothesis directly, we overexpressed LamC in glial cells using the *repo-GAL4* driver and found significant death of both glia and neurons at 10 days of age (Supplementary Fig. 2k–m).

The yes-associated protein (YAP) is a core nuclear mechanotransducer and downstream target of Hippo signaling (Fig. 1c, Supplementary Table 1), which has also been strongly linked to mechanotransduction[26, 27]. We therefore assessed YAP in Alexander disease. With double label immunofluorescence, we observed robust YAP expression in astrocyte nuclei from Alexander disease patient tissue, but not in age-matched controls (Fig. 3a). Western blot analysis confirmed the upregulation of YAP in Alexander disease patients (Fig. 3b and Supplementary Fig. 7). YAP expression was also consistently increased in the astrocytes of Alexander disease model mice, as evidenced by both immunostaining and western blot analysis (Fig. 3c,d and Supplementary Fig. 7). Yorkie (Yki), is the *Drosophila* homolog of mammalian YAP[28]. As in vertebrate models, we found that increased levels of Yki were present in the nuclei of glial cells in 20-day-old Alexander disease model flies, but not in age-matched control flies (Fig. 3e, arrows). Yki was expressed at similar levels in neurons in 20-day-old Alexander disease model flies and control flies (Fig. 3e, arrowheads). Western blot confirmed increased expression of Yki in 20-day-old Alexander disease model flies (Fig. 3f and Supplementary Fig. 7). In 1-day-old flies, Yki expression was not significantly different between Alexander disease model flies and control flies (Supplementary Fig. 3a).

To determine whether increased Yki expression led to increased Yki activity, we used three well characterized reporters of Yki activity in *Drosophila*, *ex-lacZ*[29], *cycE-lacZ*[30], and *th-GFP*[30]. All three reporters showed significant upregulation in Alexander disease model flies compared to age-matched control flies (Fig. 3g–j), indicating that Yki activity was increased in Alexander disease model flies. Thus, multiple lines of evidence support activation of YAP in Alexander disease.

To investigate whether YAP is activated through the canonical Hippo pathway in our model, we first took a genetic approach and reduced expression of the upstream negative regulator of Yki, Hippo (Hpo), using transgenic RNAi lines. Western blots confirmed increased expression of Yki in Hpo knockdown flies, with no change in GFAP[R79H] levels (Supplementary Fig. 3b,c). In addition, using double label immunofluorescence, we found the expression of phospho-Hpo and phospho-Wts, the active forms of these kinases, were reduced in the cytosol of glial cells (wrapper marks glial membranes and Repo marks the nuclei of glial cells) in Alexander disease model flies (Supplementary Fig. 3d-g, GFAP[R79H], arrows) compared to age-matched control flies (Supplementary Fig. 3d-g, Ctrl, arrowheads). Together, these data suggest canonical regulation of YAP in our model.

To determine whether Yki, a transcriptional coactivator, can regulate LamC expression, we overexpressed Yki in *Drosophila* glial cells using the *repo-GAL4* driver[31]. LamC expression was markedly increased in response to Yki expression in both control

**Fig. 3** YAP is activated in Alexander disease. **a**, **c** Double label immunofluorescence shows activated YAP in the astrocyte nuclei of a 1-year-old Alexander disease patient (AxD, arrows in **a**) and 3-month-old Alexander disease model mice (GFAP[R236H/+], arrows in **c**), but not in age-matched controls. GFAP labels astrocytes. Arrowheads (**c**) indicate weak cytoplasmic expression of YAP in wild-type control mice. **b**, **d** Western blots demonstrate marked increase of YAP expression in Alexander disease patients (AxD, **b**) and Alexander disease model mice (GFAP[R236H/+], **d**) compared to age-matched controls. $p = 0.0043$ (**b**) and 0.0286 (**d**), Mann–Whitney test. **e** Double label immunofluorescence shows activated Yki expression in the glial cell nuclei of Alexander disease model flies (GFAP[R79H], arrows), but not in age-matched controls (arrowheads). Yki is expressed similarly in the cytosol of neuronal cells in Alexander disease model flies and controls (yellow arrowheads). Repo marks glial cells. **f** Western blot demonstrates significantly increased Yki expression in Alexander disease model flies compared to age-matched controls. $N = 9$. $p = 0.0039$, Wilcoxon test. **g** β-galactosidase immunostaining shows the activation of *ex-lacZ*, a Yki activity reporter, in the glial cells of Alexander disease model flies (GFAP[R79H], arrows), but not in age-matched control. Repo marks glial cells. **h–j** Quantification of β-galactosidase-positive cells (**h**, **i**) and GFP-positive cells (**j**) shows a significant increase in Alexander disease model flies carrying Yki activity reporters *ex-lacZ* (**h**), *cycE-lacZ* (**i**) and *th-GFP* (**j**) compared to age-matched control. $N ≥ 6$ per genotype. $p = 0.0022$ (**h**), 0.0003 (**i**), and 0.0002 (**j**), Mann–Whitney test. **k**, **l** Overexpressing wild-type *yki* (**k**) or reducing *hpo* expression (**l**) in glial cells increases LamC protein levels in both control (Ctrl: *repo-GAL4/+*) and Alexander disease model flies (GFAP[R79H]: *repo-GAL4, UAS-GFAP[R79H]/+*). *$p < 0.05$, **$p < 0.01$, Wilcoxon test (**k**) and Kruskal–Wallis test (**l**). **m** Reducing *hpo* expression using transgenic RNAi lines in Alexander disease model flies markedly increased cell death measured by TUNEL analysis. $N ≥ 6$ per genotype. *$p < 0.05$, **$p < 0.01$, Kruskal–Wallis test. Flies are 20 days old in **e–m**. Scale bars are 5 microns in **a**, **e**, **g** and 10 microns in **c**. Blots are reprobed for GAPDH (**b**, **d**) or actin (**f**, **k**, **l**) to illustrate equivalent protein loading

and Alexander disease model flies (Fig. 3k and Supplementary Fig. 7). Consistent with these findings, reducing expression of the upstream negative regulator of Yki, Hippo (Hpo), using transgenic RNAi lines, also strongly increased the expression of LamC in control and Alexander disease model flies (Fig. 3l and Supplementary Fig. 7). Supporting a role for Yki upstream of

LamC, we did not find changes of Yki expression when we reduced LamC expression in control and Alexander disease model flies (Supplementary Fig. 3h). We next asked whether genetic manipulation of Hippo signaling could alter GFAP^R79H toxicity. We found that reducing Hpo expression in GFAP^R79H transgenic flies markedly increased cell death measured by

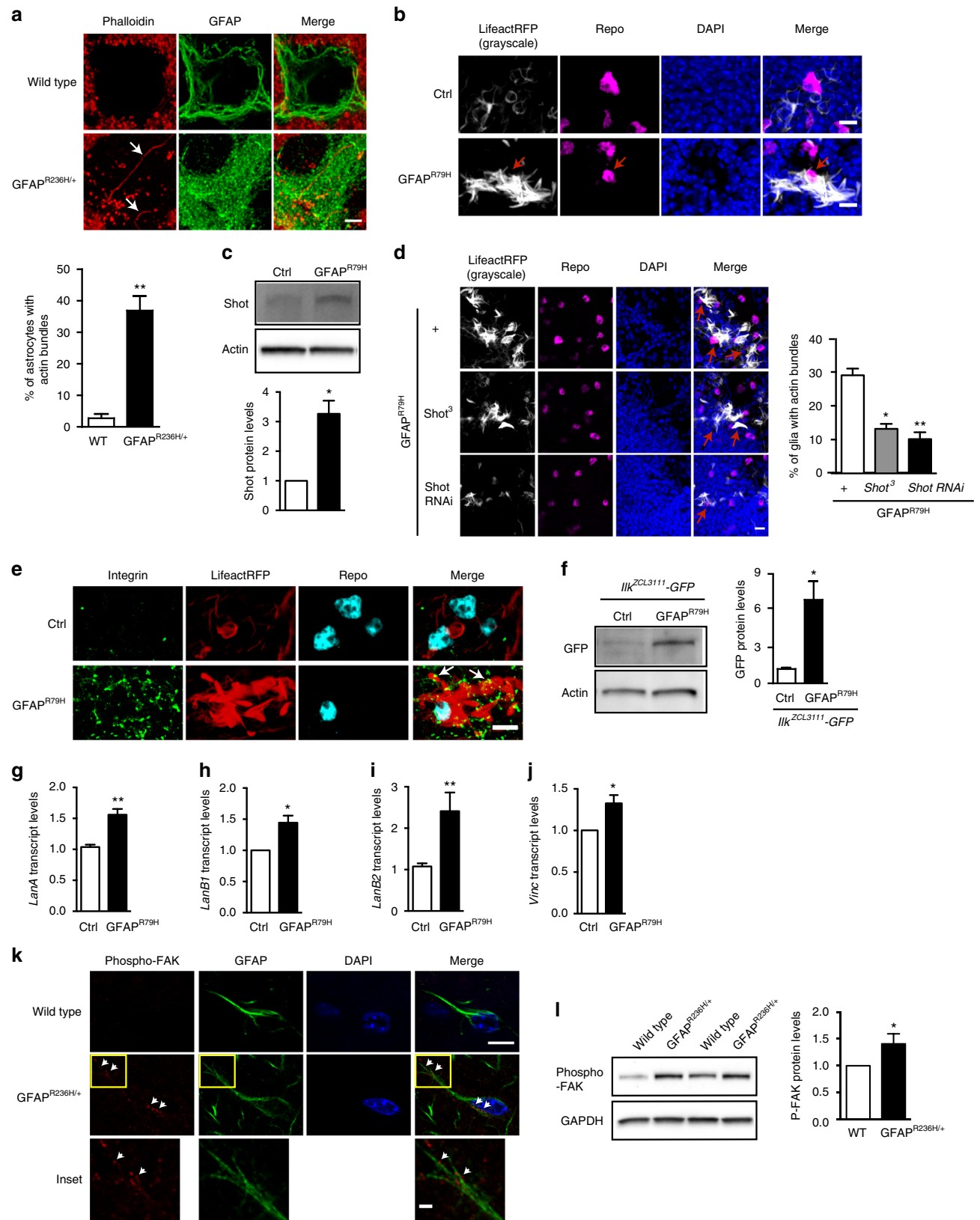

TUNEL analysis (Fig. 3m), including death of both neurons and glia by morphological analysis.

We then determined whether increasing levels of wild-type GFAP can alter mechanotransduction signaling pathways. Using double label immunofluorescence, we found significantly increased expression of lamin A and YAP in the astrocytes of human wild-type GFAP transgenic mice[12] compared to age-matched littermate controls (Supplementary Fig. 4a-b). Similarly, we found increased expression of Yki and LamC in a *Drosophila* expressing wild-type human GFAP in glial cells[21] (Supplementary Fig. 4c-d).

**Altered cytoskeletal mechanotransduction signaling.** To examine the mechanism by which increased GFAP activates Hippo signaling and increases LamC expression, we investigated cytoskeletal mechanotransducers, in particular the actin cytoskeleton and integrin focal adhesion signaling (Fig. 1b, c). Stress fibers, which are filamentous-actin (F-actin) bundles, are force-generators in the cell and mediate mechanotransduction, as well as YAP activation[26, 32]. Previous studies of primary astrocytes cultured from Alexander disease model mice demonstrated alterations in stress fiber organization[14]. To examine the actin cytoskeleton in vivo, we stained brain tissue from Alexander disease model mice with fluorescent phalloidin to label F-actin. We found that the percentage of astrocytes with visible actin bundles significantly increased in Alexander disease model mice compared to controls (Fig. 4a, Supplementary Fig. 5a and Supplementary Data 2). Similarly, by expressing LifeactRFP[33], an F-actin marker, specifically in fly glial cells using the *repo-GAL4* driver, we observed significantly increased actin bundles in the glial cells of Alexander disease model flies compared to control flies (Fig. 4b).

To probe the direct connection between GFAP and the actin cytoskeleton in Alexander disease, we focused on the spectraplakin proteins, a family of giant cytoskeletal linker proteins, which interact with F-actin and intermediate filaments[34]. Levels of the spectraplakin protein plectin are increased in Alexander disease patients and plectin co-localizes to the characteristic inclusion body of Alexander disease, the Rosenthal fiber[16]. To determine whether GFAP promotes F-actin stabilization through spectraplakin, we took advantage of the *Drosophila* model of Alexander disease. Flies have a single-spectraplakin ortholog, Short stop (Shot)[34]. Similar to increased plectin levels in Alexander disease patients, we found increased Shot expression in Alexander disease model flies compared to controls (Fig. 4c and Supplementary Fig. 7). We then reduced Shot expression using either a loss-of-function allele or a transgenic RNAi line and observed marked reductions in the number of actin bundles (Fig. 4d),

demonstrating that the spectraplakin protein Shot controls F-actin levels in GFAP[R79H] transgenic Alexander disease model flies. Western blotting confirmed reduced Shot expression in modifiers, with no change in GFAP levels (Supplementary Fig. 5b, c). Furthermore, we found increased expression of Yki and LamC when we overexpressed Shot in glial cells using the *repo-GAL4* driver, confirming that spectraplakin can control levels of mechanosensitive proteins (Supplementary Fig. 5d).

The integrin-mediated focal adhesion complex is a key transducer of mechanical force from the extracellular matrix to the actin cytoskeleton[1]. Transcriptional profiling has demonstrated increased expression of integrin genes in Alexander disease model mice[13]. Similarly, we observed increased immunostaining for one of the two *Drosophila* ß-integrin subunits (myospheroid), as well as co-localization of integrin with actin bundles in glial cells of Alexander disease model flies compared to control animals (Fig. 4e). We confirmed upregulation of the integrin pathway on western blotting by assaying levels of endogenously GFP-tagged integrin-linked kinase (Ilk)[35] (Fig. 4f and Supplementary Fig. 7). In addition to integrin proteins, we also examined extracellular matrix proteins (ECM) and intracellular focal adhesion components. Laminins are glial-secreted large heterotrimeric proteins composed of α, β, and γ subunits[36]. We found that the transcript levels of genes encoding one α chain (*LanA*), the γ subunit (*LanB1*) and the γ subunit (*LanB2*) are all significantly increased in Alexander disease model flies compared to control flies (Fig. 4g-i). Similarly, vinculin, an intracellular integrin linker protein[1], is also transcriptionally upregulated in GFAP[R79H] transgenic flies compared to age-matched controls (Fig. 4j). In addition, using double label immunofluorescence and immunoblotting, we found that the expression of phosphorylated focal adhesion kinase (FAK) protein increased significantly in the astrocytes of Alexander disease model mice compared to wild-type littermate controls (Fig. 4k, l and Supplementary Fig. 7).

**Increased brain stiffness in Alexander disease models.** The data above demonstrate alterations in mechanotransduction mediators from the transmembrane integrin-mediated focal adhesion complex to the nuclear effector YAP and lamin A/C. To address directly the mechanical properties of brain tissue in Alexander disease, we performed a series of engineering and cell biological experiments. First, we measured brain tissue elasticity in the mouse model of Alexander disease. Rotational rheometry is a well-validated and widely used technique for measurement of viscoelasticity across multiple sample types, including soft brain tissue[37]. We measured the tissue elasticity of 750 μm-thick brain sections containing the hippocampus of Alexander disease model mice and age-matched wild-type littermate control mice (Fig. 5a).

**Fig. 4** Mutant GFAP-induced cellular mechanotransduction alterations. **a** Increased F-actin bundle formation in the astrocytes of 3-month-old Alexander disease model mice (GFAP[R236H/+], arrows) compared to age-matched controls. Phalloidin staining labels F-actin. GFAP marks astrocytes. DAPI labels nuclei. Scale bar is 2 microns. $N = 4$ (wild type) and 5 (GFAP[R236H/+]). A total of 50 astrocytes per animal were used for quantification. $p = 0.0159$, Mann–Whitney test. See Supplementary Data 2 for a 3D reconstruction. **b** Abundant actin bundles in the glial cells of mutant GFAP transgenic flies (GFAP[R79H], arrows) labeled by LifeactRFP. Repo marks glial cells. Scale bar is 5 microns. $N \geq 6$ animals per genotype. $p = 0.0012$, Mann–Whitney test. **c** Western blot shows increased expression of Shot in mutant GFAP transgenic flies compared to age-matched controls. $N = 6$. $p = 0.0313$, Wilcoxon test. **d** Genetically reducing *shot* expression in Alexander disease model flies reduced the percentage of glial cells with F-actin. LifeactRFP labels F-actin. Repo marks glial cells. Scale bar is 5 microns. $N \geq 6$ per genotype. $*p < 0.05$, $**p < 0.01$, Kruskal–Wallis test. **e** Immunofluoresecence shows increased and clustered expression of integrin (arrows) along F-actin in the glial cells of Alexander disease model flies (GFAP[R79H], arrows), but not in age-matched controls. Repo marks glial cells. Scale bar is 5 microns. **f** Western blot shows increased expression of GFP in Alexander disease model flies carrying a GFP protein trap reporter for integrin-linked kinase (Ilk) compared to age-matched controls. $N = 4$. $p = 0.0286$, Mann–Whitney test. **g-j** Increased transcript levels of *LanA* (laminin α subunit), *LanB1* (laminin β subunit), *LanB2* (laminin γ subunit) and *Vinculin* (*Vinc*) in mutant GFAP transgenic flies compared to age-matched control flies. $*p < 0.05$, $**p < 0.01$, Mann–Whitney test (**g**, **i**) and Wilcoxon test (**h**, **j**). **k**, **l**, Double label immunofluorescence and western blot demonstrate increased expression phosphorylated focal adhesion kinase (phospho-FAK) in 3-month-old Alexander disease model mice (GFAP[R236H/+]) compared to age-matched wild-type littermate controls. GFAP marks astrocytes. Scale bars are 10 microns for the top two rows and 2 microns for the insets (**k**). $N = 4$, $p = 0.0221$ (**l**). Flies are 20 days old in **b-j**. Blots are reprobed for actin (**c**, **f**) or GAPDH (**l**) to illustrate equivalent protein loading

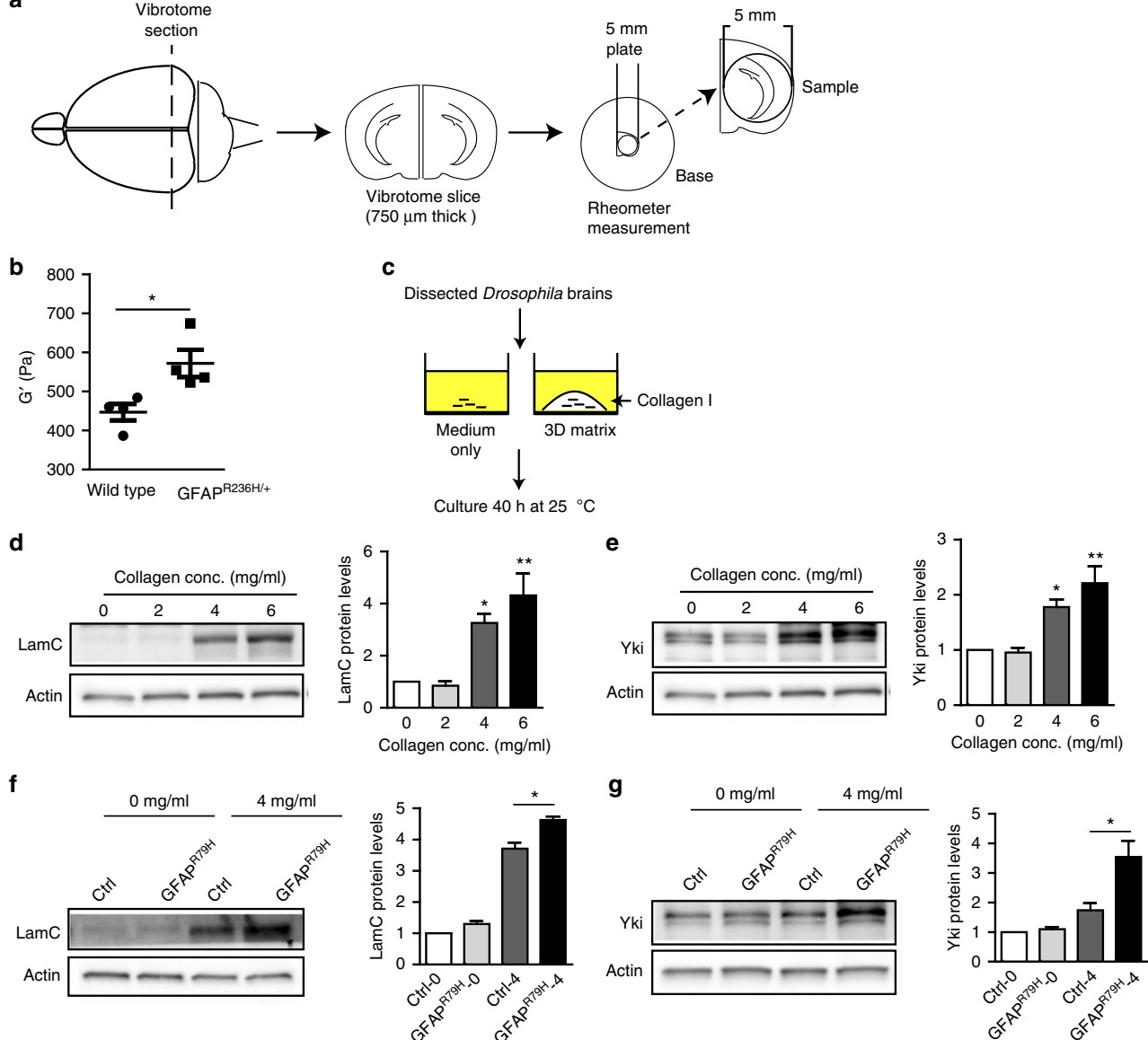

**Fig. 5** Increased brain stiffness in experimental models of Alexander disease. **a** Schematic illustration shows elasticity measurement of 750 μm-thick mouse brain slices with a rotational rheometer. **b** There is a significant increase of brain tissue stiffness in Alexander disease model mice (GFAP$^{R236H/+}$) compared to age-matched wild-type littermate controls. Mice are 4 months old. $N = 4$ per genotype. $p = 0.0286$, Mann–Whitney test. **c** Schematic illustration shows ex vivo 3D-collagen matrix culture of *Drosophila* brain. Dissected *Drosophila* brains were cultured in medium alone or in 3D-collagen matrix for 40 h at 25 °C before analysis. **d** LamC expression dose-dependently increases with collagen matrix concentrations (stiffness). Genotype: *repo-GAL4/+* (Ctrl). **e** Yki expression dose-dependently increases with collagen matrix concentrations (stiffness). Genotype: *repo-GAL4/+* (Ctrl). **f** LamC expression increases significantly more in Alexander disease model fly brains than that in control fly brains when cultured at 4 mg/ml of collagen matrix. Genotype: Ctrl: *repo-GAL4/+*; GFAP$^{R79H}$: *repo-GAL4, UAS-GFAP$^{R79H}/+$*. *$p < 0.01$. **g** Yki expression increases significantly more in Alexander disease model fly brains than that in control fly brains when cultured at 4 mg/ml of collagen matrix. Genotype: Ctrl: *repo-GAL4/+*; GFAP$^{R79H}$: *repo-GAL4, UAS-GFAP$^{R79H}/+$*. *$p < 0.01$. Flies are 1–3 days old in all panels

There was a significant increase of tissue stiffness in Alexander disease model mice ($571.7 \pm 34.74$ Pa) compared to control mice ($446.8 \pm 20.95$ Pa) (Fig. 5b and Supplementary Fig. 6a).

We next performed ex vivo three-dimensional (3D) collagen matrix culture of fly brains[38]. When we cultured *Drosophila* brains in increasing concentrations of collagen matrix of defined stiffness (56 Pa for 2 mg/ml collagen; 268 Pa for 4 mg/ml collagen and 1079 Pa for 6 mg/ml collagen as measured with a rheometer) (Fig. 5c), we found that the expression of LamC and Yki was upregulated in a dose-dependent fashion (Fig. 5d, e and Supplementary Fig. 7). Using transmission electron microscopy,

we found that mature collagen fibrils attached to neural lamella, the collagen-rich layer of extracellular matrix[39] surrounding the *Drosophila* brain, in our 3D-collagen culture conditions as shown in two representative images (Supplementary Fig. 6b, c). The brain was thus plausibly anchored directly to the culture matrix in our experiments. These findings suggest that *Drosophila* brains may be responsive to mechanical stimuli and that LamC and Yki may respond to culture matrix stiffness, although more experiments will be needed to confirm a mechanical interaction between the brain and the matrix. Next, we asked whether Alexander disease model fly brains respond to altered matrix

stiffness differently from control fly brains. To address this question, we cultured brains from control and GFAP^R79H transgenic flies in a collagen matrix stiffness (268 Pa) that promoted LamC and Yki upregulation. We found that the expression of LamC and Yki was significantly greater in Alexander disease model fly brains than in control fly brains (Fig. 5f, g and Supplementary Fig. 7). These findings suggest that GFAP^R79H expression in glia sensitized brains to altered

mechanical stress, although a stress-independent effect of increasing matrix collagen concentration cannot be excluded.

**Mechanotransduction pathways in patient astrocytes.** Patient-specific induced pluripotent stem cells (iPSCs) offer a valuable opportunity to study disease pathogenesis in appropriately differentiated cells from affected patients[40]. We therefore

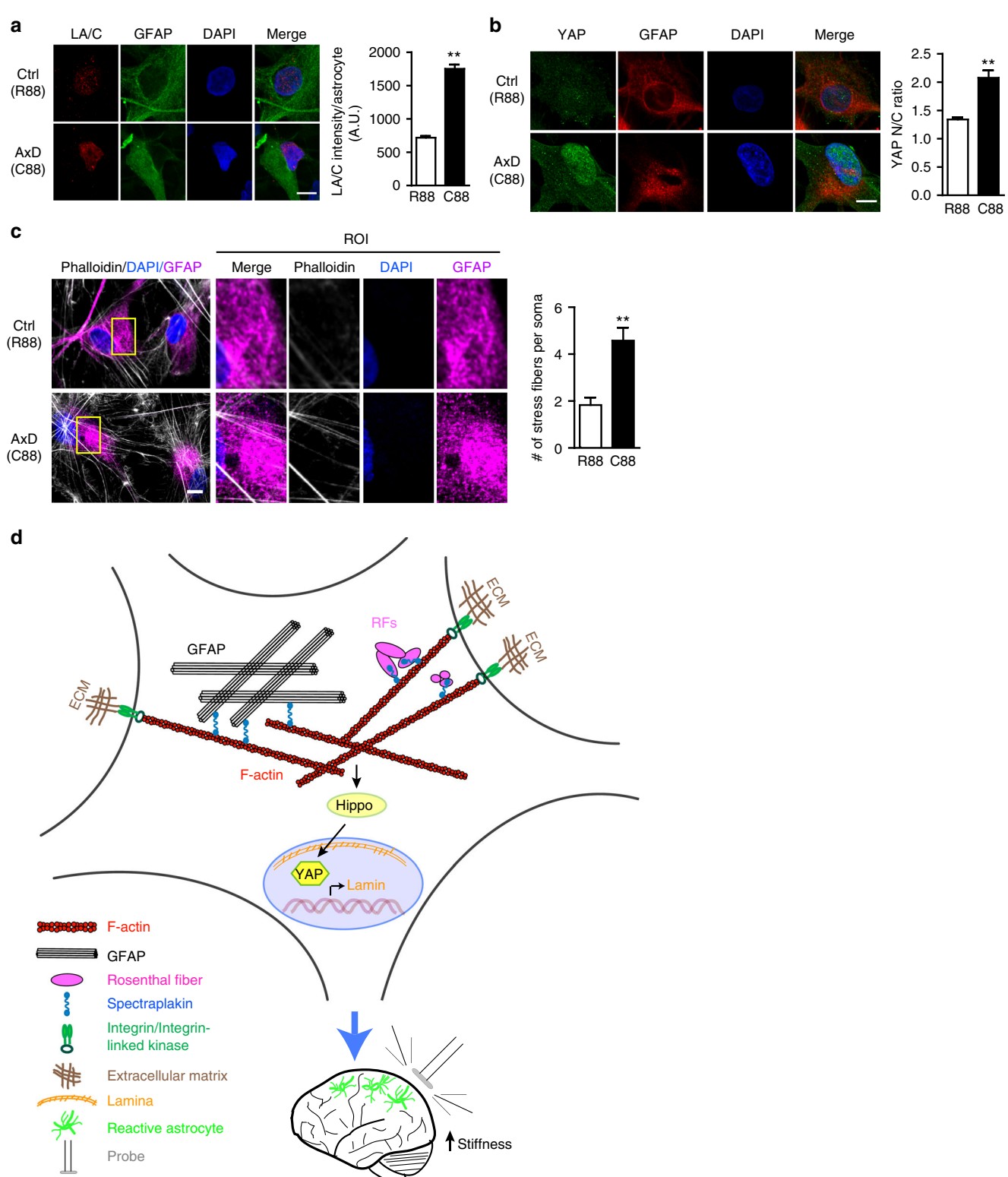

investigated key mechanosensitive markers in astrocytes differentiated from an Alexander disease iPSC line (GFAP mutation R88C) and a CRISPR/Cas9 gene corrected control line. Astrocytes were cultured on glass coverslips coated with matrigel prior to immunostaining. Consistent with results from patient brain tissue and experimental animal models, we found that the expression of A-type lamin and nuclear YAP were significantly increased in Alexander disease astrocytes compared to corrected control astrocytes (Fig. 6a, b). In addition, there were also increased numbers of stress fibers in the soma of Alexander disease astrocytes compared to corrected control cells (Fig. 6c and Supplementary Fig. 6d).

## Discussion

Here we have taken a multifaceted experimental approach integrating genetics, cell biology and biophysics in *Drosophila*, mice and humans to obtain first insights into a proposed new mechanism of disease pathogenesis in the primary astrogliopathy Alexander disease involving changes in brain stiffness and altered expression of proteins involved in mechanotransduction cascades. Our findings are consistent with a model (Fig. 6d) in which elevated expression of the glial intermediate filament GFAP promotes actin reorganization via spectraplakin proteins (plectin in mammals, Shot in *Drosophila*). In our proposed model, excessive actin stabilization directs nuclear localization and activation of the transcriptional coactivator YAP via modulation of the Hippo pathway[32]. YAP then mediates increased expression of A-type nuclear lamin. In addition, stabilization of F-actin activates integrin-mediated focal adhesion complex through "inside-out" signaling at the cell surface[1]. We propose that the concerted effects of altered mechanotransduction signaling then combine to promote increased brain stiffness (Fig. 5a–b). Interestingly, in the *Drosophila* model expression of mutant GFAP sensitizes glia to the effects of culture in increased collagen concentrations to promote further increases in LamC and Yki expression (Fig. 5c–g). These findings are intriguing given the clinical observation that several Alexander disease patients have begun to exhibit disease signs and symptoms following brain trauma[41], and are consistent with astrocyte sensitivity to substrate stiffness[42, 43].

Our findings are also consistent with prior studies demonstrating increased stiffness of astrocytes under reactive conditions[44] and with increased accumulation and maturation of cytoskeletal proteins[45]. However, not all tissue containing reactive astrocytes with elevated GFAP expression may be increased in stiffness. A decrease in tissue stiffness has been reported following spinal cord[46] and cortical[47] injury. The various factors controlling tissue stiffness in the central nervous system are likely complex. Loss-of-myelin and cyst formation[46], which are likely to be particularly relevant to stroke and other forms of localized injury, as well as location of the injury and chronicity, may all influence the degree of tissue stiffness.

Further, our work underscores the complexity of reactive astrogliosis, a spectrum of molecular, cellular and functional changes of astrocytes in response to stimuli and insults, in central nervous system physiology and pathology. Protective functions of astrogliosis have been suggested in brain ischemia of mice lacking both GFAP and vimentin (GFAP$^{-/-}$Vim$^{-/-}$)[18]. Astryocytic scar formation can promote axon regeneration after spinal cord[48]. However, studies have also shown that reactive astrogliosis can promote the development of spontaneous seizures[49], produce deficits in neuronal inhibition and perturbations in local synaptic curcuits[50], and inhibit axonal regeneration after stroke[51]. Here we propose a deleterious role of elevated GFAP in promoting activation of an astrocytic mechanotransduction signaling cascade and triggering non-cell autonomous neuronal cell death (Fig. 6d).

The exact form of GFAP that promotes F-actin stabilization in Alexander disease is unclear. A striking neuropathological feature of Alexander disease is the formation of large cytoplasmic aggregates of GFAP, termed Rosenthal fibers. The mouse and *Drosophila* models we use recapitulate Rosenthal fiber formation in affected glia. Although, Rosenthal fibers are typically associated with intermediate filaments, not thin actin filaments, when assessed by electron microscopy, proteomic and immunolocalization experiments demonstrate that the spectraplakin linker protein plectin and actin[16] are closely associated with inclusions and thus may mediate an interaction between Rosenthal fibers and the actin cytoskeleton (Fig. 6d). However, particularly in the mouse model, we observe increased F-actin in astrocytes without obvious Rosenthal fibers (Fig. 4a). Thus, GFAP assembled into intermediate filaments may interact with and stabilize actin filaments through spectraplakin (Fig. 6d). Alternatively, as with other aggregating proteins linked to human neurological diseases, GFAP can form oligomeric intermediates[52], which may exert their toxicity through actin binding and stabilization.

Increasing levels of F-actin inhibit the Hippo pathway core signaling cassette in both *Drosophila* and vertebrate systems[26, 53] and we similarly observe reduced phosphorylation of Hpo and Wts in glial cells of our Alexander disease model transgenic *Drosophila* (Supplementary Fig. 3d-g). However, multiple pathways, including cellular energy stress[54] and the unfolded protein response[55], can regulate YAP activity and the Hippo pathway. Multiple cell stress pathways are induced in Alexander disease models[52, 56, 57] and mitochondria abnormalities have been reported in Alexander disease patients[58, 59]. Thus, other mechanisms may contribute to Hippo pathway modulation in Alexander disease.

Further, Hippo signaling can regulate a number of downstream pathways. YAP/Yki is a major downstream transducer of the Hippo pathway, and has been strongly implicated in mechanotransduction[26, 53]. Although our biochemical and genetic results do argue for an important role of YAP/Yki and lamin A/C downstream of Hippo pathway signaling in mediating

**Fig. 6** Alexander disease astrocytes recapitulate key in vivo mechanosensitive markers. **a** Double label immunofluorescence and quantification demonstrate significantly increased lamin A/C expression in Alexander disease astrocytes differentiated from iPSCs (C88, bottom panel) compared to corrected control line (R88, top panel). GFAP marks astrocytes. DAPI labels nuclei. Scale bar is 10 microns. $N = 58$ (R88) and 94 (C88). $p < 0.0001$, Mann–Whitney test. **b** Double label immunofluorescence and quantification confirm increased nuclear/cytoplasmic (N/C) ratio of YAP protein in Alexander disease astrocytes (C88, bottom panel) compared to corrected control line (R88, top panel). GFAP marks astrocytes. DAPI labels nuclei. Scale bar is 10 microns. $N = 130$ (R88) and 121 (C88). $p < 0.0001$, Mann–Whitney test. **c** Double label immunofluorescence demonstrates increased stress fiber formation in Alexander disease astrocytes (C88, bottom panel) compared to corrected control line (R88, top panel). Phalloidin labels F-actin. GFAP marks astrocytes. Yellow box represents a region of interest (ROI) in the soma of each astrocyte. Scale bar is 10 microns. $N = 59$ (R88) and 50 (C88). $p < 0.0001$, Mann–Whitney test. **d** Working model of perturbed mechanotransduction homeostasis in Alexander disease. Elevated expression of GFAP promotes F-actin formation through spectraplakin protein (blue). Consequently, F-actin promote YAP nuclear translocalization and activation via the Hippo pathway. Activated YAP induces lamin expression in the nuclei. In addition, increased F-actin polymerization also activate integrin-mediated "inside-out" mechanotransduction signaling at cell membrane. Together, these alterations increase brain tissue stiffness, which in turn triggers neurodegeneration

GFAP toxicity (Fig. 3), we cannot exclude a contribution from other Hippo targets. Similarly, Hippo pathway-independent regulation of YAP/Yki can occur[26], including in mechanotransduction[27]. Our genetic data does strongly support a role for Hpo in controlling GFAP toxicity and lamin expression (Fig. 3l, m), but we cannot exclude additional Hippo-independent control of YAP/Yki.

We find increased levels of nuclear YAP/Yki in Alexander disease astrocytes (Fig. 3a) and in our experimental models of Alexander disease (Fig. 3c, e), consistent with a key nuclear role of the transcription factor[26]. Accordingly, we also observe reduced phosphorylation of Hpo and Wts (Supplementary Fig. 3d-g), which act upstream of Yki regulating nuclear localization of Yki. However, we observe increased total levels of YAP/Yki in Alexander disease (Fig. 3b, d, f), not simply redirection of YAP/Yki from the cytoplasm to the nucleus. YAP/Yki phosphorylation is known to regulate stability of the protein[26], as well as nuclear localization, providing a possible explanation for our findings. YAP levels can additionally be controlled through transcriptional mechanisms, particularly in the context of oncogenesis[60]. Further experiments will be necessary to define the role of nuclear localization, protein stabilization, and transcriptional activation in regulating YAP/Yki in Alexander disease.

Although abnormalities in A-type lamins have been associated with a number of diseases, including progeroid syndromes[61], regulation of the *LMNA* gene is still incompletely understood. There is a retinoic acid-responsive regulatory element in the promoter, which has been implicated in mechanosensitive upregulation of lamin[24]. In our studies we find that LamC is upregulated by Hippo/YAP signaling (Fig. 3k, l), raising the possibility that YAP may directly regulate lamin. Consistent with this idea, YAP binding has been documented in the upstream region of the *LMNA* gene in a glioblastoma-derived cell line[62]. Alternatively, p53 has been shown to regulate expression of A-type lamins by exerting effects at the transcriptional and posttranscriptional levels[63]. We have previously demonstrated an important role for p53 in Alexander disease pathogenesis[15], using a combination of human, mouse and *Drosophila* studies.

YAP/Yki may have other targets important in transducing and modulating the cellular response to mechanosensitive signaling. Nardone et al.[64] used ChIP-Seq and CRISPR/Cas9 gene editing to define transcriptional targets of YAP and identified multiple components of the transmembrane integrin signaling complex and extracellular matrix proteins, including collagens and laminins. Similarly, we demonstrate here that expression of integrin (Fig. 4e) and laminins (Fig. 4g-i) is upregulated in our transgenic *Drosophila* models of Alexander disease. These findings correlate well with prior work in Alexander disease model mice overexpressing wild-type human GFAP showing upregulation of integrins, collagens and laminin[13].

By providing genetic and biophysical evidence linking altered expression of proteins involved in mechanotransduction signaling cascades with behavioral and cellular toxicity in Alexander disease, our findings expand the range of possible therapeutic options available in the disorder. In particular, YAP may provide an attractive target for manipulation in patients. Given the importance of YAP in tumorigenesis, inhibitory small molecules and peptides have been developed, some of which have shown promise in mouse cancer models[65, 66].

The mechanistic and therapeutic implications of our work may extend beyond Alexander disease. Upregulation of GFAP is a common finding in diverse brain disorders, including Alzheimer's disease, Parkinson's disease, and many other disorders characterized by clinically significant tissue damage. Expansion of astrocyte cell bodies and processes by abundant eosinophilic GFAP protein is the neuropathological hallmark of a process classically termed "reactive gliosis," but increasingly recognized as functionally significant[67]. Our findings suggest that the morphological changes of gliosis may in some cases denote an alteration in mechanotransduction signaling that contributes actively to dysfunction and death of neurons and other brain cells. Although, the generalizability of our findings will require further experimental investigation, targeting the mechanotransduction pathways we outline here therapeutically may be applicable to other brain disorders.

## Methods

**Drosophila stocks and genetics.** All fly crosses were performed at 25 °C; adults were aged at 29 °C to increase transgene expression. The genotype of the *Drosophila* model of Alexander disease is: *repo-GAL4, UAS-GFAP^{R79H}/+* (GFAP^{R79H} for simplicity in the manuscript). Control is *repo-GAL4/+*. The following stocks were obtained from the Bloomington *Drosophila* Stock Center: *repo-GAL4, cycE^{05206}_lacZ, shot^3, UAS-shot-RNAi* (TRiP.HMJ233821), *UAS-shot* (29044), *Ilk^{ZCL3111}_GFP*. The following stocks were obtained from the Vienna *Drosophila* Resource Center: *UAS-yki-RNAi* (40497), *UAS-hpo-RNAi #1* (7823). Additional stocks used include *LamC^{+/-} #1* (*LamC^{Ex5}*), *LamC^{+/-} #2* (*LamC^{Ex296}*), *UAS-LamC* from L. Wallrath; *ex^{697}-lacZ* from A. Laughon; *th-GFP, UAS-myc-yki* from J. Jiang; *UAS-hpo-RNAi #2* from N. Tapon.

**Genome-scale genetic screen.** To identify pathways and molecules modifying GFAP toxicity in Alexander disease, we performed an unbiased forward genetic screen using all available transgenic RNAi lines from the Bloomington *Drosophila* Stock Center (5767 lines) and a small number of lines from the Vienna *Drosophila* Resource Center (172 lines). Among the total of 5939 lines, 94% of the genes have human homologs. Transgenic RNAi lines were crossed to an Alexander disease tester stock (genotype: *repo-GAL4, UAS-GFAP^{R79H}, UAS-CD8-PARP-Venus/TM3, Sb*), which contains the transgenic caspase reporter developed by Williams et al.[22]. Toxicity was assessed by monitoring cleavage of the reporter using an antibody (E51, Abcam) specific for cleaved human PARP. Caspase cleavage was assessed in the first 2061 transgenic RNAi lines using immunohistochemistry on tissue sections; the remainder of the lines were tested on dot blots. GFAP levels of all potential hits were examined and those that altered GFAP levels were excluded from the final modifier list. In addition, enhancers that had non-specific (GFAP^{R79H} independent) toxicity were also excluded from the final list and lines that were lethal with *repo-GAL4* (18% of lines tested) were not investigated further.

**Prize-collecting Steiner forest algorithm (PCSF).** The PCSF algorithm[23] was used to identify networks of proteins that are predicted to be involved in Alexander disease pathogenesis. Given a directed or undirected network $G(V, E, c(e), p(v))$, where $c(e)$ is the cost of an edge $e \in E$ and $p(v)$ is the prize of a node $v \in V$, we seek to maximize the prizes collected, while minimizing the cost of edges required to connect the prizes. More formally, the PCSF algorithm aims to find a forest solution $F(V_F, E_F)$ that maximize the objective function:

$$f(F) = \beta \cdot \sum_{v \in V_F} p(v) - \sum_{e \in E_F} c(e) + \omega \cdot \kappa.$$

The first term is the sum of prizes included in $F$, scaled by a model parameter $\beta$. The second term is a cost function which serves the purpose of only including a node in $F$ if the objective function is minimized. The last term allows for the inclusion of $\kappa$ trees by introducing a root node $v_0$ that is connected to every other node with a weight $\omega$. Thus, this algorithm can identify connected prize nodes, as well as "Steiner nodes" that are strongly implicated by the data.

In this study, we assigned prize values $p(v)$ to each modifier based on the magnitude with which they modify GFAP^{R79H} toxicity. Human homologs were identified through the DRSC Integrative Ortholog Prediction Tool (DIOPT). Modifiers that suppress or enhance toxicity readout were given the same positive prize values. The cost of edges $c(e)$ were assigned based our confidence in physical protein–protein interactions (PPI), as defined in prior human interaction databases. In particular, we used iRefIndex verion 13, which scores a protein interaction based on publications about that interaction, the experimental method used to detect the interaction, and the type of interaction[68].

We first ran a parameter grid search to find a network solution that included many prize nodes, and few "hub" nodes, or well-studied nodes involved in many interactions. Next, we ran a set of 100 randomization experiments, by adding noise to each edge cost. This simulates our uncertainty in the prior interaction network, which increases our confidence that nodes that appear in multiple networks are actually involved in the disease pathway. Our final PCSF network solution was the set of nodes that appeared in at least 25% of the randomization experiments, with edges defined by the PPI network. We searched the KEGG database for enriched pathways and identified nodes in the PCSF solution that belonged in these pathways. Finally, we visualized the overall network and subnetworks focusing on pathways of interest.

**Human samples**. Frozen frontal cortex from 5 controls (mean age 10 years, range 1–27 years; 2 females and 3 males) and 6 Alexander disease patients (mean age 8 years, range 1–27 years; 3 females and 3 males) were obtained from the NICHD Brain and Tissue Bank for Developmental Disorders at the University of Maryland, Baltimore, MD. GFAP mutations in the Alexander disease patients included R79C, R239C, R239H (2 cases), and K63E. All cases had typical neuropathology of Alexander disease, including multiple Rosenthal fibers. Postmortem intervals were comparable between cases and controls and were <24 h in all cases. For immunostaining analysis tissue was thawed and fixed in 4% paraformaldehyde overnight prior to paraffin embedding.

**Transgenic mice**. Four-month old (Fig. 5a, b) and 3-month old (Figs. 2c, d, 3c, d, 4a, k, l, Supplementary Fig. 2d-f, 5a) male *GFAP^R236H/+* mice[11] in the FVB/N background were used in the study. Sex- and age-matched wild-type littermates were used as controls. Three-month-old GFAP^WT mice (Tg73.7)[12] in the FVB/N background and age-matched wild-type littermates were used in Supplementary Fig. 4a,b. All procedures were approved by the Institutional Animal Care and Use Committee of the Graduate School of the University of Wisconsin-Madison and of Brigham and Women's Hospital.

**iPS cells**. Alexander disease iPS cells were derived from fibroblasts from a patient carrying the R88C mutation in GFAP[40]. The corrected control line was made via CRISPR/Cas9 gene editing (J. Jones and S. Zhang, in preparation). All cells were cultured at 37 °C with an atmosphere maintained at 5% $O_2$ and 5% $CO_2$. iPSCs were maintained on matrigel (Waisman Biomanufacturing) in E8 media. Cells were passaged every 6–7 days in the presence of ROCK inhibitor to promote cell survival. Neural induction was mostly performed via monolayer dual SMAD inhibition[69]. Neuroepithelia in the rosettes were lifted 15 days after the start of neural induction and propagated to generate astrocytes as previously described[70]. Six-month astrocyte progenitor cells were enzymatically digested with Trypsin (Gibco) and plated as single cells for maturation. Cells were plated on glass coverslips coated with matrigel prior to immunostaining. Maturation media was composed of DMEM/F12 containing 1× $N_2$, 1× NEAA, 1× glutamax, 1× pen-strep and supplemented fresh with 10 ng/mL BMP4 and 10 ng/mL CNTF. Media was changed completely every other day for 1 week before experiments.

**Immunohistochemistry, immunofluorescence, and TUNEL analysis**. For tissue sections, adult flies were fixed in formalin and embedded in paraffin. A total of 4 µm serial frontal sections were prepared through the entire fly brain and placed on a single-glass slide. In some studies, whole mount *Drosophila* brain preparations were alternatively used. Mouse and human samples were fixed in 4% paraformaldehyde, embedded in paraffin and sectioned at a thickness of 6 µm. Mouse tissue used for phalloidin staining was fixed in 4% paraformaldehyde and then cryosectioned at 6 µm thickness. Astrocytes differentiated from iPS cells were also fixed in 4% paraformaldehyde before proceeding to immunostaining.

For immunostaining, paraffin slides were processed through xylene, ethanol, and into water. Antigen retrieval by boiling in sodium citrate, pH 6.0, was performed prior to blocking. Slides were blocked in PBS containing 0.3% Triton X-100 and 2% milk for 1 h and then incubated with appropriate primary antibodies overnight.

Primary antibodies used were: anti-lamin A/C (sc6215, Santa Cruz Biotechnology) at 1:100; anti-LamC (LC28.26, Developmental Studies Hybridoma Bank) at 1:100; anti-Elav (9F8A9, Developmental Studies Hybridoma Bank) at 1:5; anti-YAP (14,074, Cell Signaling Technology) at 1:100; anti-Yki (J. Zeitlinger) at 1:500; anti-β-galactosidase (z3781, Promega) at 1:500; anti-GFP (N86/6, NeuroMab) at 1:100; acti-stain 555 phalloidin (PHDH1-A, Cytoskeleton Inc.) at 1:100; anti-GFAP (Z0334, DAKO) at 1:5000; anti-GFAP (N206/8, NeuroMab) at 1:500; anti-Repo (8D12, Developmental Studies Hybridoma Bank) at 1:5; anti-Integrin (CF.6G11, Developmental Studies Hybridoma Bank) at 1:5; anti-wrapper (10D3, Developmental Studies Hybridoma Bank) at 1:1; anti-phospho-Hpo (3681, Cell Signaling Technology) at 1:100; anti-phospho-Wts (8654, Cell Signaling Technology) at 1:250. For immunohistochemistry, biotin-conjugated secondary antibodies (1:200, Southern Biotech) and avidin-biotin-peroxidase complex (Vectastain Elite, Vector Laboratories) staining was performed using DAB (Vector Laboratories) as a chromogen. For immunofluorescence studies, appropriate Alexa fluor conjugated secondary antibodies (Alexa 488, Alexa 555 or Alexa 647) (1:200, Invitrogen) were used. All immunostaining data were replicated in at least three animals and representative images are shown.

Apoptotic cell death was visualized using terminal deoxynucleotidyl transferase biotin-dUTP nick end labeling (TUNEL) labeling according to manufacturer's instructions (TdT FragEL DNA fragmentation kit, Calbiochem), with an additional avidin-biotin-peroxidase amplification step. The number of TUNEL-positive cells was counted by examining serial frontal sections (4 µm) of the entire brains from at least six animals per genotype. In Figs. 2g and 3m, the number of TUNEL-positive cells in GFAP transgenic flies (GFAP^R79H) with LamC and Hpo modulations were compared to that in GFAP transgenic flies alone as percentage changes. Fluorescent TUNEL labeling was performed with Alexa 488 conjugated streptavidin (Invitrogen). For fluorescent double label of TUNEL with cell type-specific markers (Repo for glial cells and Elav for neuronal cells), we used Alexa 488 conjugated

streptavidin (Invitrogen) as secondary antibody for TUNEL-positive cells and Alexa 555 conjugated anti-mouse secondary antibody (Invitrogen) for Repo or Elav (Supplementary Fig. 2j,k). The percentage of neuronal cell death was quantified by dividing the number of apoptotic neuronal cells (both TUNEL- and Elav-positive cells) to the total number of apoptotic cells (all TUNEL-positive cells) (Fig. 2i and Supplementary Fig. 2m). Each data point represents Mean ± SEM.

Quantification of the number of β-gal (β-galactosidase)-positive cells for reporters *ex-lacZ*, *cycE-lacZ* and the number of GFP-positive cells for *th-GFP* (Fig. 3h–j) was performed by examining serial frontal sections (4 µm) of the entire brains from at least six animals per genotype. Each data point represents Mean ± SEM.

Quantification of LA/C intensity, YAP N/C ratio and stress fibers in iPS cells (Fig. 6a–c) was performed on confocal images (Olympus FV1000 confocal, ×63 objective for LA/C and YAP; ×100 objective for stress fibers). For stress fiber quantification, a 10 µm by 15 µm region of interest (ROI) was chosen in the soma. The number of stress fibers was counted in that ROI and statistics was done using Mann–Whitney test. Each data point represents Mean ± SEM.

**Western blots**. Human (white matter from frontal cortex) and mouse (hippocampus) samples were prepared and homogenized in RIPA buffer (50 mM Tris, 150 mM NaCl, 0.1% SDS, 0.5% deoxycholate, 1% NP40, pH 7.4, with protease inhibitor cocktail and phosphatase inhibitor cocktail, Thermo Fisher Scientific) and centrifuged at 14,000×g for 15 min at 4 °C. Supernatant was used for western blots. *Drosophila* brains were homogenized in 1× Laemmli buffer (Sigma). All samples were boiled for 10 min at 100 °C, briefly centrifuged and subjected to SDS-PAGE using 10% or 4–12% gels (Lonza). Proteins were transferred to nitrocellulose membranes (Bio-Rad), blocked in 2% milk in PBS with 0.05% Tween-20, and immunoblotted with primary antibodies. Primary antibodies used were: anti-lamin A/C (ab8984, Abcam) at 1:500; anti-LamC (LC28.26, Developmental Studies Hybridoma Bank) at 1:100; anti-GAPDH (MA5-15738, Thermo Fisher Scientific) at 1:100,000; anti-actin (JLA20, Developmental Studies Hybridoma Bank) at 1:4000; anti-Yki (J. Zeitlinger) at 1:10,000; anti-Shot (mAbRod1, Developmental Studies Hybridoma Bank) at 1:50; anti-GFP (N86/6, NeuroMab) at 1:500; anti-GFAP (Z0334, DAKO) at 1:1,000,000; anti-phospho-FAK (Tyr397) (3283, Cell Signaling Technologies) at 1:1000. The appropriate horseradish peroxidase-conjugated secondary antibody (1:20,000, Southern Biotech) was applied and signal was detected with West Femto chemiluminescent substrate (Thermo Fisher Scientific). Images were taken using a FluorChem HD2 system (ProteinSimple). All blots were repeated at least three times, and representative blots are shown in the figures. Western blots were quantified using NIH Image J software. Each data point represents Mean ± SEM.

**Behavioral analysis**. Seizure analysis was performed as previously described[21]. Briefly, flies were collected immediately after eclosion and kept at 3–5 animals per vial for 1 day, without further anesthesia before analysis. For testing, vials were mechanically stimulated on a VWR mini vortexer for 10 s at maximum speed. Seizures were defined as repetitive contractions of legs or wings, or episodes of paralysis lasting >1 s. Seizure frequency was calculated by dividing the number of flies with seizures by the total number of flies tested. Statistical significance was evaluated using the $\chi^2$-test. Each data point represents seizure frequency (% of flies with seizures).

**Mouse brain rheometry**. Brains from 4-month-old mice were sectioned at a thickness of 750 µm on a Vibrotome 1500 immediately after sacrificing. Samples were kept in cold artificial cerebrospinal fluid (0.135 M NaCl, 5.4 mm KCl, 5 mm Na-HEPES buffer, 1.8 mM $CaCl_2$, 1 mm $MgCl_2$). Brain slices (5–6 mm in diameter) containing the hippocampus were used for elasticity measurement on a strain-controlled rotational rheometer (ARES-G2, TA Instruments, DE). Measurements were performed immediately after brain sections were prepared and finished within 5 h from sacrificing. To obtain measurements, each brain slice was gently loaded between the rheometer base and 5 mm upper plate geometry. A small pressure of 5 mN was applied on tissue to assure a better contact between sample and geometry. A small oscillation was applied on samples to probe the tissue modulus. Frequency sweep from 0.1 to 10 Hz was performed with strain amplitude of 1%. A storage modulus (G′) at 0.1 Hz was used to compare brain stiffness between control and experiment groups. Experiments were performed blinded to mouse genotype. No randomization was used.

**3D-collagen matrix culture**. *Drosophila* brains were dissected in cold PBS and kept on ice before being placed in collagen matrix. The collagen matrix mixture was made according to the manufacturer's instruction (Corning Collagen I, 354249). Briefly, 1× PBS was used to dilute collagen to the desired concentration. Then collagen was pH neutralized by adding 1 N NaOH. The mixture was kept on ice until it was ready to be used. Volume of 40 µl of collagen matrix mixture per sample was added onto an ice-cold coverslip and 3–4 *Drosophila* brains were carefully embedded in the matrix. After the collagen gel solidified at 25 °C, culture medium (*Drosophila* S2 medium, 1% 10 k U/ml penicillin, 10 mg/ml streptomycin, 10% fetal bovine serum) was added and brains were cultured at 25 °C for 40 h.

**Electron microscopy**. Brains cultured in collagen matrix were fixed in 2% formaldehyde/2.5% glutaraldehyde (Polysciences). After washing in cacodylate buffer, brains were incubated in 1% osmium tetroxide (Electron Microscopy Sciences)/1.5% potassium ferrocyanide (MP Biomedicals) for 1 h, 1% uranyl acetate for 30 min, and then processed through 70, 90, and 100% ethanol solutions. Brains were then incubated in propyleneoxide for 1 h, embedded in Epon, and allowed to polymerize for 3 days at 60 °C. Thin sections were cut and examined with a Tecnai G² Spirit BioTWIN transmission electron microscope at an accelerating voltage of 80 kV.

**Confocal microscopy**. All the fluorescent images were taken on a confocal microscope (a Leica SP8× confocal microscope at Harvard NeuroDiscovery Center Enhanced Neuroimaging Core facility or an Olympus Fluoview 1000 confocal microscope at Harvard Medical School Neurobiology Imaging facility (NIH-NINDS P30NS072030). Control and experimental samples were imaged with the same laser setting. For the F-actin images in mouse tissue (Fig. 4a and Supplementary Fig. 5a), confocal images were deconvoluted using Huygens software.

**Real-time PCR**. RNA was isolated from 20-day-old *Drosophila* brains (16 brains per sample) using QIAzol (Qiagen) and reverse-transcribed with the High-Capacity cDNA Reverse Transcription kit (Applied Biosystems) according to manufacturer's instruction. Real-time PCR was performed and monitored using SYBR Green PCR Master Mix (Applied Biosystems) in a StepOnePlus Real-Time PCR system (Applied Biosystems) according to manufacturer's instructions. *Drosophila* ribosomal protein *RpL32* was used as a control. Each data point represents Mean ± SEM.

Primers:

*LanA*: 5′-CCAGGGGCATGAAATTCAATGAAGTC-3′; 5′-GGTCTTGCCAT AATCCGTGGACT-3′

*LanB1*: 5′-CTCGCCGGAGAGATTCTGCA-3′; 5′-TTGTACGGATCATGCTT GGTCTCC-3′

*LanB2*: 5′-GGCCACAGAAATGTCTGCCAG-3′; 5′-CTGCTCACCACAGGTA TTAGTTGACTC-3′

*Vinc*: 5′-GCATCTTTCCAACCAGAATGCCGA-3′; 5′-CGATCAAGATGGGC TGGTTGGC-3′

*Rpl32*: 5′-GACCATCCGCCCAGCATAC-3′; 5′-CGGCGACGCACTCTGTT-3′

**Statistical analysis**. Statistical analysis was performed using GraphPad Prism 6 software. Each data point represents Mean ± SEM. For comparison of 2 groups, parametric ($t$-test) or non-parametric (Mann–Whitney test or Wilcoxon test) tests were done where appropriate. For comparison of three groups or more, parametric One-way ANOVA or non-parametric Kruskal–Wallis/Friedman tests were done where appropriate. The $\chi^2$-test was used for seizure behavior analysis. $N$ and $p$-values are indicated in figure legends. $p < 0.05$ was considered statistically significant. Sample size was decided based on the minimal number required for statistical significance or previous experience in the laboratory/literature. Variance within each group was not calculated, but assumed to be similar. Data for all experiments were collected and processed randomly, but no formal randomization was carried out. Data collection and analysis were not performed blind given the nature of the experiments performed.

**Data availability**. All data are available from the authors upon request.

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

## Acknowledgements

Fly stocks obtained from the Bloomington *Drosophila* Stock Center (NIH P40OD018537), the Vienna *Drosophila* Resource Center, L. Wallrath, A. Laughon, J. Jiang and N. Tapon were used in this study. We thank the Transgenic RNAi Project (TRiP) at the Harvard Medical School (NIH-NIGMS R01GM084947) for making transgenic RNAi stocks. Monoclonal antibodies were obtained from the Developmental Studies Hybridoma Bank developed under the auspices of the NICHD and maintained by the University of Iowa, Department of Biology, Iowa City, IA 52242, and the UC Davis/NIH NeuroMab Facility. J. Zeitlinger kindly provided the *Drosophila* Yki antibody. Human tissue was obtained from the NICHD Brain and Tissue Bank for Developmental Disorders at the University of Maryland, Baltimore, MD. This work was supported by the NIH-NICHD P01HD076892, the Intellectual and Developmental Disabilities Research Centers at Boston Children's Hospital (NIH-NICHD U54HD090255) and the Waisman Center (NIH-NICHD U54HD090256), the National Science Foundation (DMR-1310266), the Harvard Materials Research Science and Engineering Center (DMR-1420570), NIH P01HL120839-03, and NIH P01GM096971.

## Author contributions

L.W. and M.B.F. designed the study and wrote the manuscript. L.W., J.X., T.L.H., and J.J. performed experiments and edited the manuscript. J.L., D.W., S.-C.Z., A.M., and E.F. analyzed data and edited the manuscript. M.B.F., S.C.-Z., and A.M. obtained funding for the research.

## Additional information

**Competing interests:** The authors declare no competing interests.

