## [Peer Review File · Nature Communications]

Reviewers' expertise:

Reviewer #1: mechanosensitivity in CNS diseases;

Reviewer #2: Alexander disease;

Reviewer #3: gliosis in CNS diseases.

Reviewers' comments:

Reviewer #1 (Remarks to the Author):

In this manuscript, Wang et al. investigate the relationship of misregulated GFAP expression in glial cells and mechanotransduction pathways. They exploit a variety of experimental systems related to Alexander disease and different experimental approaches to show an upregulation of YAP/Yki and lamin A/C / LamC, key players in mechanotransduction in different systems, in the disease. Particularly the link between GFAP, spectraplakins and actin is intriguing. This work thus contributes important novel findings to the field that will be relevant for a wide readership.

The paper is generally well written, figures are good, and the experiments appear sound. However, I have a couple of concerns and questions that should be addressed:

1. A number of statements are not backed up by data and should be toned down. Much of what the authors show are correlations, causality can usually not be concluded. For example, the authors state 'Excessive actin stabilization directs nuclear localization and activation of the transcriptional coactivator YAP via modulation of the Hippo pathway.' All the authors show are correlations, causality has not been shown. Also, modulation of the Hippo pathway has not been shown (see below). 'The concerted effects of altered mechanotransduction signaling combine to promote increased brain stiffness (Fig. 5a-d)' Again, no causality has been shown, only a correlation (if the brain stiffness results are reliable, see below). Furthermore, the authors conclude that an 'increased expression of LamC triggers neurodegeneration'. This statement is not backed up by their data, as all that is currently shown is a correlation between cell death and LamC expression in the disease model. In order to validate this statement, LamC should also be upregulated in controls.
2. Similarly, the authors conclude that LamC expression is increased because of an increased activity of Yki. While the presented data are convincing and in support of this hypothesis, they do not rule out that changes in LamC (which might be a more direct consequence of changes in GFAP) might vice versa lead to changes in Yki (which, if easily doable, might be worth checking). Hence, this might be a chicken and egg problem, which should be more critically discussed.
3. As mentioned above, I find the proposed link between spectraplakins and mechanotransduction exciting. What is currently missing, however, is to show a direct link between spectraplakins / Shot, Yap / Yki and lamin A/C / LamC. For example, in the flies with reduced shot expression, Yki and LamC levels should be quantified to directly show that these proteins are linked.
4. Is the Hippo pathway really activated? In Dupont et al. 2011 it is shown that mechanotransduction may lead to YAP translocation and activation independent of the Hippo pathway. If what is observed is indeed due to mechanical signals, what evidence do the authors have that Hippo is involved?
5. YAP does not seem to be expressed in control cells (Fig. 3). If mechanical signals would lead to the accumulation of YAP in nuclei of astrocytes in cells derived from diseased patients or animals, I would expect it to be expressed also in controls but rather being confined to the cytoplasm, as it is described in many other mechanotransduction studies? It should be discussed why YAP expression or translation

is / are upregulated rather than it being translocated into the nucleus.

6. Fig 4 is called 'GFAP-induced cellular mechanotransduction alterations'. How do the authors know that the alterations are caused (induced) by changes in GFAP? Again, it seems to me that what is shown is a correlation.

7. I am not convinced about the mechanics measurements of the brain. First, the samples used for shear rheology did not fit the used geometry. The resulting sample overflow will lead to wrong results (see, for example, Ewoldt, R.H., M.T. Johnston, L.M. Caretta, "Experimental challenges of shear rheology: how to avoid bad data," in: S. Spagnolie (Editor), *Complex Fluids in Biological Systems*, Springer (2015)). Samples need to be trimmed to match the geometry size in order to obtain reliable results. The velocity-based approach is rather indirect and has not been validated with a sample of known properties. I am not sure if it is really brain elasticity that is measured here or rather its viscosity (which is rate-dependent)? Control measurements need to be done (at least on mouse brains once they have been properly measured by shear rheology).

8. It is not clear to me if and how embedding whole brains in a 3D collagen gel stiffens the tissue? It should be shown that the brains in these conditions indeed become stiffer. If brains do stiffen, is this due to coupling to the gels or is collagen being incorporated into the tissue before it fully polymerizes? This should be discussed. A correlation of collagen concentration in the surrounding matrix and LamC and Yki expression in cells within the brain does not automatically 'demonstrate that Drosophila brains are indeed responsive to mechanical stimuli and that LamC and Yki respond directly to culture matrix stiffness'. As far as I understand, most of the cells in the brain are not in contact with the collagen matrix? How would they feel the stiffness of the collagen matrix? Can the authors exclude effects of chemical signals? This statement needs to be backed up by data or toned down. Finally, causality between brain stiffening and disease progression could only be shown if brains were softened, and levels of LamC and Yki as well as symptoms would go down.

9. While this study focused on mechanotransduction pathways, iPS cells seem to have been cultured on standard tissue culture plastics rather than on compliant substrates of defined stiffness matching the mechanics of the healthy and diseased brain, which has become a standard technique in many labs (although no information on the cultures is provided in the methods). I'm a bit puzzled about that, this system would have provided a straight-forward way to test the impact of environmental stiffness on single cell behavior and thus to test causality?

Minor points:

1. Actin bundles are not automatically stress fibers, which should be under tension and thus straight. I cannot see stress fibers in Fig. 4. I would recommend calling these structures actin bundles. (the difference in actin organization is obvious though)
2. The discussion is a bit short and might benefit from an extension. For example, the authors should discuss papers in which a correlation between substrate stiffness and gliosis has been shown. Also, the authors should cite Lu et al 2011 (FASEB J), in which a correlation between increased GFAP expression by glial cells has been shown to correlate with an increase in glial stiffness.
3. The authors have exclusively used statistical tests assuming the data to be normally distributed. Has this assumption been tested? This should be stated. If data are not normally distributed, non-parametric test should be used instead.

Reviewer #2 (Remarks to the Author):

This is an interesting and novel study that raises an issue of brain "stiffness" associated with astrogliosis, in this case illustrated by a variety of changes in the CNS of Alexander disease, an

"astrogliopathy" caused by mutations in the GFAP gene. The findings could reveal important, common changes in astrocytes in gliosis in a wide variety of neurological disorders and so will be of interest to both glial cell biologists and investigators of many neurological diseases.

There are several "stories" in this manuscript. One theme deals with the increased mechanical stiffness of Alexander brain tissues and astrocytes. Another theme suggests that this increased stiffness results in increased cell death. Another theme makes progress in unraveling some of the molecular changes that may underlie mechanical changes in these cells and tissues. The authors have made progress on each of these issues, some more in depth than others. One could argue, however, that each of these requires a fuller inquiry to reach clearer conclusions. Thus, the paper could influence thinking about the effects of astrogliosis, but each of these themes is not well enough established.

Individual Comments:

1. Is it the "stiffness" per se that results in more cell death in the fly model? Reducing Hippo levels increased cell death, but did not alter GFAP levels. How should these results be interpreted? Hippo intersects with many other signaling systems, so an increased LamC will not be the only change, and changes in the transcription of other genes may have significant effects on cell survival or death. Furthermore, GFAP levels do not change. Given that increased GFAP levels are the characteristic feature of this disease, the implication is that some effect downstream of the increase in GFAP is significantly increased or decreased. What might that be?
2. Lamin A protein levels are increased in Alexander disease astrocytes in mouse and human (Figure 2), where the protein appears to localize to the periphery of the nucleus. Levels of LaminA are also increased in white matter in several human Alexander patients, although not all. What is missing here is a localization of LaminA in the human and mouse Alexander brains, rather than the astrocytes in culture. Presumably LaminA is found in many cell types, and although the implication here is that its increase is specific to astrocytes, it may be that other cell types also show increases. Changes in other cell types might also contribute to the increases in stiffness of the tissue.
3. The above discussion is also germane to the cellular localization of the Yap increase.
4. Decreasing LamC expression in *Drosophila* reduced cell death, measured by TUNEL labeling (Figure 2g). However, the authors do not show what kind(s) of cells these are. Are they neurons? Are they the pathological astrocytes? The answer is important. The statement that "Non-cell autonomous neuronal loss is a prevalent feature of Alexander disease, which is recapitulated in our fly model" suggests that there is less neuronal death after decreasing LamC, but this is not proven. Similarly, the title that increased stiffness triggers "neurodegeneration" in Alexander disease implies that it is the stiffness that plays a causal role in neurodegeneration.
5. The LamC experiment shows, once again, the value of *Drosophila* models in revealing mechanisms of neurodegenerative diseases. However, the authors do not present evidence that there is appreciable cell (neuronal or other) death in the Alexander mice. Reducing LamC in Alexander mice seems out of the bounds of this manuscript, but even so, if there is little cell death in the mice, it may be likely that reducing LamC will not produce less cell death. The authors could co-culture the Alexander astrocytes with neurons to ask if there is significant cell death and then decrease LamC in astrocytes.
6. Figure 4 displays data on stress fiber presence in astrocytes. In the culture system, very few of the wt astrocytes show stress fibers. Astrocytes, like many other cells in culture, can show stress fibers

under the right conditions, particularly if they are flattened against the substrate. Were the cell densities and cell spreading identical for the wt and Alexander astrocytes? The illustration in 4a shows an astrocyte with 1 or 2 stress fibers. The authors might choose a different example, showing a cell with multiple fibers. The illustrations in Figure 6 and in the Extended Data Figure 4 do show stress fibers in both control and iPS human astrocytes. Was the data in the Extended Figure 4 also quantitated? In Figure 6 the data are presented as number of stress fibers per cell, different from the data in Figure 4. Is there any reason for this?

7. Stress fibers also reflect interactions with substrate. Perhaps the Alexander astrocytes have more focal adhesions. If so, how could that increase be related to the molecular changes the authors have found. For example, is focal adhesion kinase more heavily phosphorylated in these astrocytes? Or are there increases in focal adhesion proteins like vinculin?

8. The authors mention that integrin genes are upregulated in Alexander disease. It might be interesting to mention which ones, particularly ones related to focal adhesions and mechano-sensory signaling, such as α V β 1 (see Nardone, Nature Communications, 2017, May 15).

9. The data from the 3D model with increasing collagen concentrations implies that stiffness the only variable/effect to take into account when collagen concentration is increased? Could the presence of collagen have other effects? For example, there could be changes in levels of cell surface ECM receptors.

10. In the brain stiffness test, the authors only look at the hippocampus. One assumes that this is the case because of a higher degree of gliosis there. It would be interesting in the future to look at cortex, or subcortical white matter, if possible.

11. Figure 6 is a diagram that illustrates some of the findings and speculations in this work. One of the important questions is whether the ECM surrounding astrocytes changes, and if so, how those changes are transmitted into astrocytes to establish more actin cables. What is known about changes in ECM in any of the Alexander models?

12. The diagram in Figure 6 shows Rosenthal fibers linked to F-actin cables. Is there evidence for this? Rosenthal fibers are seen in the immediate proximity of intermediate filaments and appear linked to them.

13. The authors might consider that members of the plakin/plectin family of linker proteins can also interact with cell nuclei through the nesprin family of proteins. In Alexander disease, there might be alterations in nuclear mechanotransduction.

Reviewer #3 (Remarks to the Author):

This manuscript reports on the effects of expressing a gain-of-function mutant form of GFAP associated with Alexander disease (AxD) in various animal models. AxD is a rare disease that is generally caused by spontaneous mutations in a single allele of the GFAP that gives rise to full-length forms of mutant GFAP (see ref 7 in this paper). AxD-mutant-GFAP causes severe tissue pathology and disease by gain-of-function, because homozygous loss of GFAP causes little or no effect, whereas expression a single copy of AxD-mutant-GFAP in transgenic mice replicates human AxD pathology and causes related pathologies in flies (ref 7). I elaborate on these details because they are important in understanding my comments and concerns. All of the results presented here are caused by over-expressing a highly toxic mutant-GFAP, but the authors inappropriately extend and generalize their

findings to implying similar effects would be caused by over-expressing normal GFAP as happens in almost all CNS injuries and diseases.

The manuscript presents a number of interesting findings pertaining to AxD-mutant-GFAP. The premise of the study is logically based on evaluation of a genetic screen of changes exhibited by drosophila astrocytes expressing AxD-mutant-GFAP that revealed multiple nodes involved in mechanotransduction pathways. What follows is an elegant series of experiments testing the effects of AxD-mutant-GFAP expression on specific molecules in pathways that regulate mechanotransduction in multiple models, including transgenic flies, transgenic mice, human postmortem tissue and astrocytes derived from ESCs from AxD patients. In all of these models the authors present compelling evidence that expression of AxD-mutant-GFAP alters proteins critically involved in mechanotransduction pathways, as well as altering mechanotransduction itself. The experimental design is logical and appropriate. The experiments appeared well conducted and properly controlled. The basic data presented are convincing, and in this regard I have no specific comments or concerns.

However, I found the results to be inappropriately over-interpreted and over-generalized in a manner that is not scientifically rigorous and is not in any way justified by the either the data presented or vast array of literature from many laboratories over the past twenty years. The context of the paper is presented in a misleading manner right from the first sentence of the abstract. The authors appear intent on trying to use this study which deals only with a highly toxic mutant form of GFAP to further their notion that upregulation of normal GFAP during reactive gliosis is somehow toxic, and that by extension, normal gliosis in many disease contexts is toxic via the specific mechanism of altered mechanotransduction. This is not in any way justified by the results of this study. They present no data of any kind in this paper to support this contention, and the paper in its present form should not be published here or anywhere. To do so would be misleading, and damaging to the field, because it would mislead non-experts and encourage them to propagate this unfounded notion. That said, the basic data in the paper are of high quality and the findings on mechanotransduction are novel and would be interesting, so long as the authors make perfectly clear that their results pertain at this point only to a mutant form of GFAP that is toxic and not to normal GFAP. A more balanced introduction and discussion of the consequences of GFAP expression by normal reactive astrocytes, and a more balance representation of the vast amount that is already known about the roles of reactive astrocytes are essential.

Major concerns:

1. The first two sentences of the abstract are inaccurate and misleading. Regarding the first sentence, there are over twenty years of genetically targeted loss-of-function studies ranging from deletion of GFAP itself to selective deletion of transcriptional regulators, as well as more recent transcriptome evaluations under different experimental manipulations, that together have identified a number of specific functions of reactive astrocytes and underlying mechanisms, including immune regulatory functions, phagocytic functions, tissue protective functions and neurotoxic functions that may be related to clearing neurons that are severely damaged for example by axotomy. To say that "Gliosis, ... is of unknown functional significance" is inaccurate and glib, and sounds like a self-serving attempt to try and make the results here seem more important than they are. It would equally inappropriate to start the abstract by saying that the function of upregulation of GFAP in reactive astrocytes is not known, even if this statement is true, because this study is not about the upregulation of normal GFAP. It is about the upregulation of a highly toxic mutant form of GFAP. For this reason, the second sentence in the abstract is equally misleading. The authors cannot say that their findings in any way show (or even suggest) that misregulation of GFAP leads to changes in mechanoconduction that cause tissue damage. The can at best say that mutation of GFAP as happens in AxD does so. The abstract needs to be focused on the effects of AxD-mutant-GFAP on mechanotransduction and not generalize to normal GFAP or to other neurological disorders.

2. If a revised version of this manuscript is considered for publication, the authors must alter their terminology throughout the paper to use the term mutant-GFAP instead of merely GFAP when referring to all of their models and the effects induced in those models. For example, on page 3 in the introduction, they do not examine a GFAP-knockin models but rather AxD-mutant-GFAP-knockin model in mice and flies. The difference is critical. It is terribly misleading to not make this perfectly clear to the reader. There are dozens of places in the paper where the term "GFAP" is used that must be changed to "mutant-GFAP" or "AxD-mutant-GFAP". Not to do so would be terribly misleading. For example in the first paragraph on the results on page 4, the authors refer to "suppressors and enhancers of GFAP toxicity". This is completely inappropriate. There is no known toxicity of GFAP. What is examined here is the toxicity of "AxD-mutant-GFAP". This must be made clear every time the term "GFAP" is used throughout the paper.

3. In the second paragraph of the introduction on page 3, the authors need to make clear that AxD is caused by a gain-of-function mutation and that mutation of a single allele is sufficient to cause disease, as described in their own review article (ref # 7). This is critical for non-expert readers.

4. It is clear from their review article (ref # 7) that these authors collectively favor a concept of GFAP toxicity, whereby accumulation of normal GFAP beyond a certain point is toxic, but they have never presented any compelling evidence in support of such a hypothesis. This notion is based largely on a single study of mice over-expressing GFAP conducted twenty years ago and not published on since then. Only truly massive levels of GFAP over-expression to levels never seen in any form of reactive gliosis were toxic. This kind of toxicity might be said of many other proteins. There is no evidence that expression of GFAP at levels encountered after injury and disease might be toxic. It is not appropriate for the authors here to try and further their notion of GFAP toxicity by disguising the fact that they looked only at mutant-GFAP in this study.

5. If the authors want to claim that increased expression of wild-type GFAP alters mechanotransduction then they need to conduct studies by over-expressing WT GFAP. Again it is not appropriate for the authors here to try and further this notion by disguising the fact that they looked only at mutant-GFAP in this study.

6. In their discussion, the authors selectively cherry pick the literature to cite the only paper that reports any form of mechanistic evidence for a toxic astrocyte phenotype (ref # 49). In doing so, they are being disingenuous. One has to wonder if the present authors even read more than the title of that paper? That paper, in contrast to this one, is written in a scholarly and balanced manner. It refers equally to protective (A2) and toxic (A1) functions of reactive astrocytes and discusses literature providing evidence for both. Indeed, evidence from that lab (Barres lab) shows that protective (A2) astrocytes express GFAP levels equal to or greater than toxic (A1) ones (see Zamanian et al paper from the Barres lab). Indeed, there is ample evidence that reactive astrocytes that have clearly been demonstrated to be protective express equal levels of GFAP to those that are implicated as being toxic. The authors should discuss this. In addition, there are multiple studies in which deletion of GFAP results in more, not less, tissue damage after stroke or in mice with Alzheimer's pathology. (see Li L et al. (2008) Protective role of reactive astrocytes in brain ischemia. *J Cereb Blood Flow Metab* 28:468-481. – and Kraft et al, (2013) Attenuating astrocyte activation accelerates plaque pathogenesis in APP/PS1 mice. *FASEB J* 27:187-198.) The authors here need to read the literature better and present a more balanced and scholarly introduction and discussion. The last paragraph of the discussion is truly a one-sided over-reach that is not at all justified by the data in this study.

7. Last but not least, the authors should check the literature better on tissue stiffness after CNS injury. A paper recently published in this journal presents results that have bearing on the work presented here. Moeendarbary et al (2017) The soft mechanical signature of glial scars in the central

nervous system. Nature communications 8:14787. (I am not a co-author). This study uses atomic force microscopy and convincingly shows that glial scars in vivo were unexpectedly softer (not stiffer) than uninjured tissue. Glial scars are formed largely by astrocytes that express among the highest levels of GFAP encountered after any form of pathology. This observation does not fit with the contention by the present authors that upregulated expression of normal GFAP massively increases tissue stiffness, and suggests that instead it is the A β D-mutant-GFAP that is causing this. The authors here need to discuss this point as well, and not ignore it.

Response to Reviewers

The Reviewer's comments are reproduced in full (*italics*) followed by our responses (plain text).

Reviewers' expertise:

Reviewer #1: mechanosensitivity in CNS diseases;
Reviewer #2: Alexander disease;
Reviewer #3: gliosis in CNS diseases.

Reviewers' comments:

Reviewer #1 (Remarks to the Author):

In this manuscript, Wang et al. investigate the relationship of misregulated GFAP expression in glial cells and mechanotransduction pathways. They exploit a variety of experimental systems related to Alexander disease and different experimental approaches to show an upregulation of YAP/Yki and lamin A/C / LamC, key players in mechanotransduction in different systems, in the disease. Particularly the link between GFAP, spectraplakins and actin is intriguing. This work thus contributes important novel findings to the field that will be relevant for a wide readership.

We appreciated the positive evaluation of the Reviewer.

The paper is generally well written, figures are good, and the experiments appear sound. However, I have a couple of concerns and questions that should be addressed:

1. A number of statements are not backed up by data and should be toned down. Much of what the authors show are correlations, causality can usually not be concluded. For example, the authors state 'Excessive actin stabilization directs nuclear localization and activation of the transcriptional coactivator YAP via modulation of the Hippo pathway.' All the authors show are correlations, causality has not been shown. Also, modulation of the Hippo pathway has not been shown (see below). 'The concerted effects of altered mechanotransduction signaling combine to promote increased brain stiffness (Fig. 5a-d)' Again, no causality has been shown, only a correlation (if the brain stiffness results are reliable, see below).

The Reviewer raises important points regarding causality in our studies and we have addressed each concern in full, as detailed below. The exact quotes provided by the Reviewer are taken from the discussion of our proposed model. We have revised our Discussion section to clarify instances in which we discuss our model vs. our experimental findings.

Furthermore, the authors conclude that an 'increased expression of LamC triggers neurodegeneration'. This statement is not backed up by their data, as all that is currently shown is a correlation between cell death and LamC expression in the disease model. In order to validate this statement, LamC should also be upregulated in controls.

As suggested, we have increased LamC expression and show that overexpression

results in both glial and neuronal cell death (Supplementary Fig. 2k-m).

2. Similarly, the authors conclude that LamC expression is increased because of an increased activity of Yki. While the presented data are convincing and in support of this hypothesis, they do not rule out that changes in LamC (which might be a more direct consequence of changes in GFAP) might vice versa lead to changes in Yki (which, if easily doable, might be worth checking). Hence, this might be a chicken and egg problem, which should be more critically discussed.

To address this possibility, we reduced LamC expression using two separate loss of function alleles in both control and GFAP transgenic flies. We did not see changes in Yki levels following modulation of LamC expression (Supplementary Fig. 3h).

3. As mentioned above, I find the proposed link between spectraplakine and mechanotransduction exciting. What is currently missing, however, is to show a direct link between spectraplakine / Shot, Yap / Yki and lamin A/C / LamC. For example, in the flies with reduced shot expression, Yki and LamC levels should be quantified to directly show that these proteins are linked.

To demonstrate a more direct link between shot and Yki/LamC we overexpressed shot (*Drosophila* spectraplakine) in fly glial cells and observed increased expression of both Yki and LamC (Supplementary Fig. 5d).

4. Is the Hippo pathway really activated? In Dupont et al. 2011 it is shown that mechanotransduction may lead to YAP translocation and activation independent of the Hippo pathway. If what is observed is indeed due to mechanical signals, what evidence do the authors have that Hippo is involved?

We believe that Hpo plays an important role in regulating mechanotransduction based on results of genetic knockdown of Hpo (Fig. 3l,m). In addition, we now demonstrate altered levels of phospho-Hpo and phospho-Wts (phosphorylation of Hpo and its downstream target Wts/Lats would lead to depression of Yki) in glial cells of GFAP transgenic flies (Supplementary Fig. 3d-g), to provide complementary evidence for regulation of the Hippo pathway. However, as suggested by the Reviewer, we cannot exclude an additional Hippo-independent effect on YAP, which we now discuss in the manuscript.

5. YAP does not seem to be expressed in control cells (Fig. 3). If mechanical signals would lead to the accumulation of YAP in nuclei of astrocytes in cells derived from diseased patients or animals, I would expect it to be expressed also in controls but rather being confined to the cytoplasm, as it is described in many other mechanotransduction studies? It should be discussed why YAP expression or translation is / are upregulated rather than it being translocated into the nucleus.

The original images provided did not adequately demonstrate the cytoplasmic signal for YAP, and new photos are provided in Fig. 3, which illustrate cytoplasmic YAP more clearly. It is, however, correct that we observe an increase in total YAP/Yki levels. Regulation of YAP is well documented both at the level of nuclear localization as well as through degradation of the protein. Thus, YAP/Yki degradation may be inhibited in GFAP transgenic animals. Alternatively, there may be increased transcription of YAP/Yki. These points are now addressed in the Discussion.

6. Fig 4 is called 'GFAP-induced cellular mechanotransduction alterations'. How do the authors know that the alterations are caused (induced) by changes in GFAP? Again, it seems to me that what is shown is a correlation.

The changes we observe are present in GFAP transgenic animals, and not in age- and sex-matched controls. Thus, these alterations are induced by GFAP. We further provide evidence that GFAP mediates effects on mechanotransduction via the F-actin cytoskeleton and Hippo pathway as diagrammed in Figure 6d. While it is possible that other pathways also contribute to altered mechanotransduction downstream of GFAP in transgenic animals, the activation of mechanotransduction pathways specifically in GFAP transgenic animals, and not controls, provides strong evidence that GFAP induces the alterations.

7. I am not convinced about the mechanics measurements of the brain. First, the samples used for shear rheology did not fit the used geometry. The resulting sample overfill will lead to wrong results (see, for example, Ewoldt, R.H., M.T. Johnston, L.M. Caretta, "Experimental challenges of shear rheology: how to avoid bad data," in: S. Spagnolie (Editor), *Complex Fluids in Biological Systems*, Springer (2015)). Samples need to be trimmed to match the geometry size in order to obtain reliable results.

The Reviewer is correct regarding the importance of matching sample and geometry. We should have provided more methodological detail on this important point, and we have now done so. In fact, the size of the mouse brain sample (approximately 5 – 6 mm in diameter) closely matched the 5 mm upper plate of the rheometer. We realize that our diagram may have been misleading on this point, and we have now altered Fig. 5a to more clearly represent our actual experimental conditions.

According to the rheology measurement calculation, the modulus is based on the strain input and the torque feedback from the sample. Here we used a strain controlled rheometer, which means the strain is 100% accurate. The only error potentially contributing to the modulus calculation is from the torque (M, see the function below), which is calculated by integrating the shear stress (τ) across the whole plate.

$$M = \int_0^{2\pi} \int_0^R r^2 \tau_{\phi\theta} \Big|_{(\pi/2)} dr d\phi = \frac{2\pi R^3}{3} \tau_{\phi\theta} \Big|_{(\pi/2)}$$

Since any modest excess sample has no constraint on the top and rests loosely on the bottom geometry due to lack of pressure from the top surface, it is unlikely to contribute significantly to the stress inside the sample. As a result, the torque error coming from the small excess sample is likely to be very small. Further, we note that all our samples were the same size. Any small error attributable to sample size should be the same for control and GFAP transgenic animals.

Reference:

Middleman, S. *Rheology: Principles, measurements, and applications* by C. Macosko, VCH Publishers, 1994, 550pp. *AIChE J.*, **41**, 2344 (1995).

The velocity-based approach is rather indirect and has not been validated with a sample of known properties. I am not sure if it is really brain elasticity that is measured here or

rather its viscosity (which is rate-dependent)? Control measurements need to be done (at least on mouse brains once they have been properly measured by shear rheology).

We do not believe that our measurements reflect viscosity. Although the samples deformed upon laser application, deformation ended promptly when the laser force was removed. If our measurement reflected velocity, the sample would have continued to deform with a given speed. Although the PIV method is commonly used in velocity measurements, here we use PIV to compare the **deformation amplitude, not velocity**. We have modified Fig. 5c,d to clarify this point.

8. It is not clear to me if and how embedding whole brains in a 3D collagen gel stiffens the tissue? It should be shown that the brains in these conditions indeed become stiffer. If brains do stiffen, is this due to coupling to the gels or is collagen being incorporated into the tissue before it fully polymerizes? This should be discussed. A correlation of collagen concentration in the surrounding matrix and LamC and Yki expression in cells within the brain does not automatically 'demonstrate that Drosophila brains are indeed responsive to mechanical stimuli and that LamC and Yki respond directly to culture matrix stiffness'. As far as I understand, most of the cells in the brain are not in contact with the collagen matrix? How would they feel the stiffness of the collagen matrix? Can the authors exclude effects of chemical signals? This statement needs to be backed up by data or toned down. Finally, causality between brain stiffening and disease progression could only be shown if brains were softened, and levels of LamC and Yki as well as symptoms would go down.

We apologize for not explaining our experiment design more clearly. The outer surface of the fly brain is comprised of a collagen-rich neural lamella (Supplementary Fig. 6b,c, nl). We reasoned that collagen in the matrix would attach to the neural lamella and thus physically anchor the brain to the matrix. To test our hypothesis directly we have performed transmission electron microscopy on *Drosophila* brains embedded in collagen under the conditions used in Fig. 5. In Supplementary Fig. 6b,c arrows point to typical collagen fibrils in the matrix. The insets show collagen fibrils attached to the neural lamella of the fly brain (yellow arrowheads). Of course, we cannot completely exclude direct activation of conventional chemical signaling cascades, but this caveat applies to cells cultured in matrices of different stiffness as well.

9. While this study focused on mechanotransduction pathways, iPSC cells seem to have been cultured on standard tissue culture plastics rather than on compliant substrates of defined stiffness matching the mechanics of the healthy and diseased brain, which has become a standard technique in many labs (although no information on the cultures is provided in the methods). I'm a bit puzzled about that, this system would have provided a straight-forward way to test the impact of environmental stiffness on single cell behavior and thus to test causality?

iPSCs were plated on matrigel, not directly on tissue culture plastic, as we detail in the revised manuscript. We agree that culturing iPSCs in matrix of different stiffness would be an interesting experiment. Unfortunately, given the significant amount of time required to differentiate cells into astrocytes (at least 6 months) and the limited number of Alexander disease and genotypically corrected control cells available to us currently, we were unable to perform the experiment suggested. We note that we did demonstrate altered sensitivity of Alexander disease model glial cells to matrix of different stiffness by performing the organ culture experiments presented in Fig. 5e-j.

Minor points:

1. *Actin bundles are not automatically stress fibers, which should be under tension and thus straight. I cannot see stress fibers in Fig. 4. I would recommend calling these structures actin bundles. (the difference in actin organization is obvious though)*

We thank the Reviewer for bringing this point to our attention. We have altered our wording appropriately.

2. *The discussion is a bit short and might benefit from an extension. For example, the authors should discuss papers in which a correlation between substrate stiffness and gliosis has been shown. Also, the authors should cite Lu et al 2011 (FASEB J), in which a correlation between increased GFAP expression by glial cells has been shown to correlate with an increase in glial stiffness.*

We have expanded our Discussion as suggested by the Reviewer, including citing the reference given.

3. *The authors have exclusively used statistical tests assuming the data to be normally distributed. Has this assumption been tested? This should be stated. If data are not normally distributed, non-parametric test should be used instead.*

We have provided additional information on our statistical methods, and have used nonparametric tests where appropriate.

Reviewer #2 (Remarks to the Author):

This is an interesting and novel study that raises an issue of brain “stiffness” associated with astrogliosis, in this case illustrated by a variety of changes in the CNS of Alexander disease, an “astrogliopathy” caused by mutations in the GFAP gene. The findings could reveal important, common changes in astrocytes in gliosis in a wide variety of neurological disorders and so will be of interest to both glial cell biologists and investigators of many neurological diseases.

There are several “stories” in this manuscript. One theme deals with the increased mechanical stiffness of Alexander brain tissues and astrocytes. Another theme suggests that this increased stiffness results in increased cell death. Another theme makes progress in unraveling some of the molecular changes that may underlie mechanical changes in these cells and tissues. The authors have made progress on each of these issues, some more in depth than others. One could argue, however, that each of these requires a fuller inquiry to reach clearer conclusions. Thus, the paper could influence thinking about the effects of astrogliosis, but each of these themes is not well enough established.

Individual Comments:

1. *Is it the “stiffness” per se that results in more cell death in the fly model? Reducing Hippo levels increased cell death, but did not alter GFAP levels. How should these results be interpreted? Hippo intersects with many other signaling systems, so an increased LamC will not be the only change, and changes in the transcription of other*

genes may have significant effects on cell survival or death. Furthermore, GFAP levels do not change. Given that increased GFAP levels are the characteristic feature of this disease, the implication is that some effect downstream of the increase in GFAP is significantly increased or decreased. What might that be?

In our *Drosophila* model we express GFAP through a heterologous promoter system (GAL4/UAS) so that we can direct our experimental attention specifically to the events downstream of GFAP that mediate cellular toxicity, which we believe include an important component of mechanical stress signaling. We cannot rule out effects of the Hippo pathway that are independent of LamC, which we now state in the Discussion. To provide additional evidence regarding the involvement of LamC in our model we have included several new experiments in our revised manuscript. First, we have increased LamC expression and show that overexpression results in both glial and neuronal cell death (Supplementary Fig. 2k-m). To demonstrate a direct link between Shot and LamC we overexpressed Shot (*Drosophila* spectraplakins) in fly glial cells and observed increased expression of LamC (Supplementary Fig. 5d), and of Yki as well. We have also addressed the possibility that LamC might reciprocally regulate Yki. We reduced LamC expression using two separate loss of function alleles in both control and GFAP transgenic flies. We did not see changes in Yki levels following modulation of LamC expression (Supplementary Fig. 3h). These findings, taken together, strengthen the evidence for a key role of LamC downstream of the Hippo pathway and provide additional details regarding regulation of the signaling cascade in our models.

2. Lamin A in protein levels are increased in Alexander disease astrocytes in mouse and human (Figure 2), where the protein appears to localize to the periphery of the nucleus. Levels of LaminA are also increased in white matter in several human alexander patients, although not all. What is missing here is a localization of LaminA in the human and mouse Alexander brains, rather than the astrocytes in culture. Presumably LaminA is found in many cell types, and although the implication here is that its increase is specific to astrocytes, it may be that other cells types also show increases. Changes in other cells types might also contribute to the increases in stiffness of the tissue.

3. The above discussion is also germane to the cellular localization of the Yap increase.

The data in Fig. 2a,b (Lamin A) and Fig. 3a,b (YAP) are from human postmortem brain, and in Fig. 2c,d (Lamin A) and 3c,d (YAP) are from mouse brain, not from cultured astrocytes. We have ensured that the figure legends clarify this important point. To address the cell type specific regulation of Lamin A and YAP we performed double label immunofluorescent analyses with cell type specific markers (Supplementary Fig. 2a-c and Supplementary Fig. 2d-f). In Alexander disease patient tissue, we found that the expression of lamin A in oligodendrocytes and neurons were similar to that in control tissue (Supplementary Fig. 2a-b). We did not observe expression of lamin A in microglial cells under our experimental conditions (Supplementary Fig. 2c). We found similar results in the mouse model of Alexander disease (Supplementary Fig. 2d-f). In patient tissue and our mouse model, we found strong YAP immunostaining only in Alexander disease models. There was also weak staining for YAP in the cytoplasm of astrocytes from age-matched control mouse tissue (Fig 3c, top). In the *Drosophila* model of Alexander disease, we observed expression of Yki in the cytosol of neuronal cells, but levels of neuronal Yki were not different between GFAP transgenic flies and controls (Fig. 3e. yellow arrowheads).

4. Decreasing LamC expression in Drosophila reduced cell death, measured by TUNEL labeling (Figure 2g). However, the authors do not show what kind(s) of cells these are. Are they neurons? Are they the pathological astrocytes? The answer is important. The statement that “Non-cell autonomous neuronal loss is a prevalent feature of Alexander disease, which is recapitulated in our fly model” suggests that there is less neuronal death after decreasing LamC, but this is not proven. Similarly, the title that increased stiffness triggers “neurodegeneration” in Alexander disease implies that it is the stiffness that plays a causal role in neurodegeneration.

We agree that cell type specific analysis of cell death is needed to substantiate our statements. Thus, we have used double label immunofluorescence with cell type specific markers to identify the dying cells as neurons or astrocyte-like glia. The Methods section has now been expanded to clarify our experimental approach. Thus, in Fig. 2i we supplement our analysis of the total number of TUNEL-positive cells (Fig. 2g) with cell type specific data to demonstrate that neuronal death is rescued by reduction of LamC.

5. The LamC experiment shows, once again, the value of Drosophila models in revealing mechanisms of neurodegenerative diseases. However, the authors do not present evidence that there is appreciable cell (neuronal or other) death in the Alexander mice. Reducing LamC in Alexander mice seems out of the bounds of this manuscript, but even so, if there is little cell death in the mice, it may be likely that reducing LamC will not produce less cell death. The authors could co-culture the Alexander astrocytes with neurons to ask if there is significant cell death and then decrease LamC in astrocytes.

We agree that the Alexander disease model mice we use, while an excellent genetic mimic of the human disease, have more modest neuropathology. Although co-culture would be one way to address the issue of downstream neuropathology in a vertebrate system, given the somewhat artificial nature of the culture experiment we favor an in vivo approach. Accordingly, we are working hard to characterize a rat model of Alexander disease, which preliminary data suggest will replicate a wider range of neuropathology than the mouse model.

6. Figure 4 displays data on stress fiber presence in astrocytes. In the culture system, very few of the wt astrocytes show stress fibers. Astrocytes, like many other cells in culture, can show stress fibers under the right conditions, particularly if they are flattened against the substrate. Were the cell densities and cell spreading identical for the wt and Alexander astrocytes? The illustration in 4a shows an astrocyte with 1 or 2 stress fibers. The authors might choose a different example, showing a cell with multiple fibers. The illustrations in Figure 6 and in the Extended Data Figure 4 do show stress fibers in both control and iPS human astrocytes. Was the data in the Extended Figure 4 also quantitated? In Figure 6 the data are presented as number of stress fibers per cell, different from the data in Figure 4. Is there any reason for this?

The reviewer is correct that stress fibers are easily demonstrated in cultured astrocytes. However, in vivo, as presented in Fig. 4a, demonstrable actin bundles (see comment from Reviewer #1 above) are infrequent. In contrast, Fig. 6 presents data from cultured iPS cells. Cell density and culture conditions were identical for the Alexander disease iPS cells and corrected control lines. Supplementary Fig. 6d (formerly Supplementary Fig. 4) presents individual, unmerged channels from the composite shown in Fig. 6c to allow clear comparison of stress fibers. Quantification is presented in Fig. 6c. We have revised the figure legend to clarify this point.

7. Stress fibers also reflect interactions with substrate. Perhaps the Alexander astrocytes have more focal adhesions. If so, how could that increase be related to the molecular changes the authors have found. For example, is focal adhesion kinase more heavily phosphorylated in these astrocytes? Or are there increases in focal adhesion proteins like vinculin?

The Reviewer makes a good point. As suggested, we examined phosphorylated focal adhesion kinase protein (phospho-FAK) in the mouse model of Alexander disease using a phospho-specific FAK antibody. Both immunofluorescence and western blot demonstrate increased expression of phospho-FAK in the astrocytes of Alexander disease model mice compared to age-matched control mice (Fig. 4k,l). Using quantitative qRT-PCR, we found increased *Vinculin* (*Vinc*) mRNA in GFAP transgenic flies compared to age-matched controls (Fig. 4j).

8. The authors mention that integrin genes are upregulated in Alexander disease. It might be interesting to mention which ones, particularly ones related to focal adhesions and mechano-sensory signaling, such as *alphaVbeta1* (see Nardone, *Nature Communications*, 2017, May 15).

As suggested, we have now specified that expression of integrins beta-1 and beta-5 (*Itgb1* and *Itgb5*) have been shown to be elevated in a transgenic mouse model of Alexander disease based on overexpression of wild type human GFAP (Hagemann, et al., 2005), and have discussed the relevant findings of the paper by Nardone and colleagues. In our *Drosophila* GFAP transgenic flies, we demonstrated that β PS integrin has increased expression compared to age-matched control flies (Fig. 4e).

Reference:

Hagemann, T.L. *et al.* Gene expression analysis in mice with elevated glial fibrillary acidic protein and Rosenthal fibers reveals a stress response followed by glial activation and neuronal dysfunction. *Hum. Mol. Genet.* **14**, 2443-2458 (2005).

9. The data from the 3D model with increasing collagen concentrations implies that stiffness the only variable/effect to take into account when collagen concentration is increased? Could the presence of collagen have other effects? For example, there could be changes in levels of cell surface ECM receptors.

We realize that we should have explained our experimental design more clearly. The outer surface of the fly brain is comprised of a collagen-rich neural lamella (Supplementary Fig. 6b,c, nl). We reasoned that collagen in the matrix would attach to the neural lamella and thus physically anchor the brain to the matrix. To test our hypothesis directly we have performed transmission electron microscopy on *Drosophila* brains embedded in collagen under the conditions used in Fig. 5. In Supplementary Fig. 6b,c arrows point to typical collagen fibrils in the matrix. The insets show collagen fibrils attached to the neural lamella of the fly brain (yellow arrowheads). We therefore believe that the primary effect of increasing matrix stiffness is mechanical. Although we cannot completely exclude direct activation of conventional chemical signaling cascades, this caveat also applies to cells cultured in matrix of different stiffness.

10. In the brain stiffness test, the authors only look at the hippocampus. One assumes that this is the case because of a higher degree of gliosis there. It would be interesting in

the future to look at cortex, or subcortical white matter, if possible.

We agree. As the reviewer indicates, we focused on the hippocampus because gliosis was robust in the hippocampus and because clear increases in Lamin A and YAP were observed. In the future it would be of interest to look in other areas of the brain.

11. Figure 6 is a diagram that illustrates some of the findings and speculations in this work. One of the important questions is whether the ECM surrounding astrocytes changes, and if so, how those changes are transmitted into astrocytes to establish more actin cables. What is known about changes in ECM in any of the Alexander models?

To investigate changes in extracellular matrix molecules in GFAP transgenic flies we performed quantitative qRT-PCR. We found that the transcript levels of one laminin α subunit (*LanA*), the β subunit (*LanB1*) and the γ subunit (*LanB2*) were significantly increased in Alexander disease model flies compared to age-matched control flies (Fig. 4g-i). Similarly, multiple collagen genes and *Lama4* (encoding laminin α -4) are upregulated in a transgenic mouse model of Alexander disease based on overexpression of wild type human GFAP (Hagemann, et al., 2005).

Reference:

Hagemann, T.L. *et al.* Gene expression analysis in mice with elevated glial fibrillary acidic protein and Rosenthal fibers reveals a stress response followed by glial activation and neuronal dysfunction. *Hum. Mol. Genet.* **14**, 2443-2458 (2005).

12. The diagram in Figure 6 shows Rosenthal fibers linked to F-actin cables. Is there evidence for this? Rosenthal fibers are seen in the immediate proximity of intermediate filaments and appear linked to them.

Although we are not aware of electron microscopic evidence for an association of thin filaments with Rosenthal fibers, we intended to indicate in our schematic that the exact species of GFAP (filaments, oligomers, Rosenthal fibers) interacting with the F-actin cytoskeleton in disease states remains to be defined. Strong evidence (both immunostaining and proteomic) does support the presence of spectraplakin in Rosenthal fibers, and spectraplakin can link the actin cytoskeleton and intermediate filaments. We have expanded our discussion on this important point.

13. The authors might consider that members of the plakin/plectin family of linker proteins can also interact with cell nuclei through the nesprin family of proteins. In Alexander disease, there might be alterations in nuclear mechanotransduction.

The reviewer makes an excellent point, and we plan to address nuclear alterations downstream of cytosolic mechanotransduction signaling pathways in subsequent experimental efforts.

Reviewer #3 (Remarks to the Author):

This manuscript reports on the effects of expressing a gain-of-function mutant form of GFAP associated with Alexander disease (AxD) in various animal models. AxD is a rare disease that is generally caused by spontaneous mutations in a single allele of the

GFAP that gives rise to full-length forms of mutant GFAP (see ref 7 in this paper). AxD-mutant-GFAP causes severe tissue pathology and disease by gain-of-function, because homozygous loss of GFAP causes little or no effect, whereas expression a single copy of AxD-mutant-GFAP in transgenic mice replicates human AxD pathology and causes related pathologies in flies (ref 7). I elaborate on these details because they are important in understanding my comments and concerns. All of the results presented here are caused by over-expressing a highly toxic mutant-GFAP, but the authors inappropriately extend and generalize their findings to implying similar effects would be caused by over-expressing normal GFAP as happens in almost all CNS injuries and diseases.

The manuscript presents a number of interesting findings pertaining to AxD-mutant-GFAP. The premise of the study is logically based on evaluation of a genetic screen of changes exhibited by drosophila astrocytes expressing AxD-mutant-GFAP that revealed multiple nodes involved in mechanotransduction pathways. What follows is an elegant series of experiments testing the effects of AxD-mutant-GFAP expression on specific molecules in pathways that regulate mechanotransduction in multiple models, including transgenic flies, transgenic mice, human postmortem tissue and astrocytes derived from ESCs from AxD patients. In all of these models the authors present compelling evidence that expression of AxD-mutant-GFAP alters proteins critically involved in mechanotransduction pathways, as well as altering mechanotransduction itself. The experimental design is logical and appropriate. The experiments appeared well conducted and properly controlled. The basic data presented are convincing, and in this regard I have no specific comments or concerns.

However, I found the results to be inappropriately over-interpreted and over-generalized in a manner that is not scientifically rigorous and is not in any way justified by the either the data presented or vast array of literature from many laboratories over the past twenty years. The context of the paper is presented in a misleading manner right from the first sentence of the abstract. The authors appear intent on trying to use this study which deals only with a highly toxic mutant form of GFAP to further their notion that upregulation of normal GFAP during reactive gliosis is somehow toxic, and that by extension, normal gliosis in many disease contexts is toxic via the specific mechanism of altered mechanotransduction. This is not in any way justified by the results of this study. They present no data of any kind in this paper to support this contention, and the paper in its present form should not be published here or anywhere. To do so would be misleading, and damaging to the field, because it would mislead non-experts and encourage them to propagate this unfounded notion. That said, the basic data in the paper are of high quality and the findings on mechanotransduction are novel and would be interesting, so long as the authors make perfectly clear that their results pertain at this point only to a mutant form of GFAP that is toxic and not to normal GFAP. A more balanced introduction and discussion of the consequences of GFAP expression by normal reactive astrocytes, and a more balance representation of the vast amount that is already known about the roles of reactive astrocytes are essential.

We appreciate the assessment of the Reviewer that our work is “elegant,” “well conducted and properly controlled,” and “convincing.” We further apologize for inadvertently presenting an unbalanced view of research on glial influences on neurological diseases and neglecting to reference and discuss important prior contributions appropriately. We realize from comments made by each of the Reviewers that our work required a more extensive and nuanced discussion section to place our findings properly in the context of prior published data and discuss caveats and alternative interpretations in proper detail. We have thus substantially revised the

manuscript to address the Reviewers' comments.

Major concerns:

1. *The first two sentences of the abstract are inaccurate and misleading. Regarding the first sentence, there are over twenty years of genetically targeted loss-of-function studies ranging from deletion of GFAP itself to selective deletion of transcriptional regulators, as well as more recent transcriptome evaluations under different experimental manipulations, that together have identified a number of specific functions of reactive astrocytes and underlying mechanisms, including immune regulatory functions, phagocytic functions, tissue protective functions and neurotoxic functions that may be related to clearing neurons that are severely damaged for example by axotomy. To say that "Gliosis, ... is of unknown functional significance" is inaccurate and glib, and sounds like a self-serving attempt to try and make the results here seem more important than they are. It would equally inappropriate to start the abstract by saying that the function of upregulation of GFAP in reactive astrocytes is not known, even if this statement is true, because this study is not about the upregulation of normal GFAP. It is about the upregulation of a highly toxic mutant form of GFAP. For this reason, the second sentence in the abstract is equally misleading. The authors cannot say that their findings in any way show (or even suggest) that misregulation of GFAP leads to changes in mechanotransduction that cause tissue damage. They can at best say that mutation of GFAP as happens in AxD does so. The abstract needs to be focused on the effects of AxD-mutant-GFAP on mechanotransduction and not generalize to normal GFAP or to other neurological disorders.*

We have previously demonstrated that overexpression of wild type human GFAP in mice and flies replicates key features of Alexander disease (Messing et al., 1998; Hagemann, et al., 2005; Wang et al., 2011; Wang et al, 2015). We now show that the mechanosensitive markers lamin A and YAP are increased in astrocytes of mice expressing wild type human GFAP (Supplementary Fig. 4a,b). Similarly, flies expressing wild type human GFAP have increased LamC and Yki (Supplementary Fig. 4c,d). These changes, though statistically significant, are less robust in flies expressing wild type human GFAP compared to flies expressing R79H mutant human GFAP, correlating with increased in vivo toxicity of the mutant protein (Wang et al., 2011 and Wang et al., 2015).

References:

- Messing, A. *et al.* Fatal encephalopathy with astrocyte inclusions in GFAP transgenic mice. *Am. J. Pathol.* **152**, 391–398 (1998).
- Hagemann, T.L. *et al.* Gene expression analysis in mice with elevated glial fibrillary acidic protein and Rosenthal fibers reveals a stress response followed by glial activation and neuronal dysfunction. *Hum. Mol. Genet.* **14**, 2443-2458 (2005).
- Wang, L., Colodner, K.J. & Feany, M.B. Protein misfolding and oxidative stress promote glial-mediated neurodegeneration in an Alexander disease model. *J. Neurosci.* **31**, 2868-2877 (2011).
- Wang, L. *et al.* Nitric oxide mediates glia-induced neurodegeneration in Alexander disease. *Nat. Commun.* **6**, 8966 (2015).

2. *If a revised version of this manuscript is considered for publication, the authors must alter their terminology throughout the paper to use the term mutant-GFAP instead of merely GFAP when referring to all of their models and the effects induced in those*

models. For example, on page 3 in the introduction, they do not examine a GFAP-knockin models but rather AxD-mutant-GFAP-knockin model in mice and flies. The difference is critical. It is terribly misleading to not make this perfectly clear to the reader. There are dozens of places in the paper where the term “GFAP” is used that must be changed to “mutant-GFAP” or “AxD-mutant-GFAP”. Not to do so would be terribly misleading. For example in the first paragraph on the results on page 4, the authors refer to “suppressors and enhancers of GFAP toxicity”. This is completely inappropriate. There is no known toxicity of GFAP. What is examined here is the toxicity of “AxD-mutant-GFAP”. This must be made clear every time the term “GFAP” is used throughout the paper.

As discussed above, we now include expression of wild type GFAP in our manuscript. We have reviewed our figures carefully to ensure that the type of GFAP expressed is denoted in each panel so there can be no misunderstanding regarding the species of GFAP studied.

3. In the second paragraph of the introduction on page 3, the authors need to make clear that AxD is caused by a gain-of-function mutation and that mutation of a single allele is sufficient to cause disease, as described in their own review article (ref # 7). This is critical for non-expert readers.

We have made the indicated change.

4. It is clear from their review article (ref # 7) that these authors collectively favor a concept of GFAP toxicity, whereby accumulation of normal GFAP beyond a certain point is toxic, but they have never presented any compelling evidence in support of such a hypothesis. This notion is based largely on a single study of mice over-expressing GFAP conducted twenty years ago and not published on since then. Only truly massive levels of GFAP over-expression to levels never seen in any form of reactive gliosis were toxic. This kind of toxicity might be said of many other proteins. There is no evidence that expression of GFAP at levels encountered after injury and disease might be toxic. It is not appropriate for the authors here to try and further their notion of GFAP toxicity by disguising the fact that they looked only at mutant-GFAP in this study.

Our collaborative group has previously described a wild type GFAP transgenic mouse model (GFAP^{WT}) made by overexpressing human GFAP using the endogenous promoter (Messing et al., 1998). We have subsequently published extensively on the wild type GFAP transgenic model (Hagemann et al., 2005, 2006, 2009, 2012; Cho et al., 2009; Meisingset et al., 2010; Cunningham et al., 2013; Jany et al., 2014; Walker et al., 2014; Wang et al., 2015; Heaven et al., 2016). The GFAP protein level in the GFAP^{WT} model is approximately 3-5-fold greater than in control mice at 14 days old and 20-fold in 12-14 months old mice. For comparison, in one study reactive astrocytes isolated from two mouse injury models (ischemic stroke and neuroinflammation), GFAP transcript levels increase 7.5-fold compared to sham astrocytes (Zamanian J. et al., 2012).

The wild type GFAP transgenic *Drosophila* model (*repo-GAL4, UAS-GFAP^{WT}/+*; GFAP^{WT} for simplicity) was generated by expressing human wild type human GFAP in glial cells using the *repo-GAL4* driver (Wang et al., 2011). The protein levels of GFAP in GFAP^{WT} transgenic flies and mutant-GFAP transgenic flies (GFAP^{R79H}) are equivalent (Wang et al., 2011).

References:

- Cho, W. & Messing, A. Properties of astrocytes cultured from GFAP-over-expressing and GFAP mutant mice. *Exp. Cell Res.* **315**, 1260-1272 (2009).
- Cunningham, R., Jany, P., Messing, A. & Li, L. Protein changes in immunodepleted cerebrospinal fluid from transgenic mouse models of Alexander disease detected using mass spectrometry. *J. Proteome Res.* **12**, 719-728 (2013).
- Hagemann, T.L. *et al.* Gene expression analysis in mice with elevated glial fibrillary acidic protein and Rosenthal fibers reveals a stress response followed by glial activation and neuronal dysfunction. *Hum. Mol. Genet.* **14**, 2443-2458 (2005).
- Hagemann, T.L., Connor, J.X. & Messing, A. Alexander disease-associated glial fibrillary acidic protein mutations in mice induce Rosenthal fiber formation and a white matter stress response. *J. Neurosci.* **26**, 11162-11173 (2006).
- Hagemann, T.L., Boelens, W., Wawrousek, E. & Messing, A. Suppression of GFAP toxicity by α B-crystallin in mouse models of Alexander disease. *Hum. Mol. Genet.* **18**, 1190-1199 (2009).
- Hagemann, T.L., Jobe, E.M. & Messing, A. Genetic ablation of Nrf2/antioxidant response pathway in Alexander disease mice reduces hippocampal gliosis but does not impact survival. *PLoS ONE* **7**, e37304 (2012).
- Heaven, M.R. *et al.* The composition of Rosenthal fibers, the protein aggregate hallmark of Alexander disease. *J. Proteome Res.* **55**, 2265-2282 (2016).
- Jany, P.L., Hagemann, T.L. & Messing, A. GFAP expression as an indicator of disease severity in mouse models of Alexander disease. *ASN Neuro.* **5**, e00109 (2013).
- Meisingset, T.W., Risa, Ø., Brenner, M., Messing, A. & Sonnewald, U. Alteration of glial-neuronal metabolic interactions in a mouse model of Alexander disease. *Glia.* **58**, 1228-1234 (2010).
- Messing, A. *et al.* Fatal encephalopathy with astrocyte inclusions in GFAP transgenic mice. *Am. J. Pathol.* **152**, 391-398 (1998).
- Walker, A.K. *et al.* Astrocytic TDP-43 pathology in Alexander disease. *J. Neurosci.* **34**, 6448-6558 (2014).
- Wang, L., Colodner, K.J. & Feany, M.B. Protein misfolding and oxidative stress promote glial-mediated neurodegeneration in an Alexander disease model. *J. Neurosci.* **31**, 2868-2877 (2011).
- Wang, L. *et al.* Nitric oxide mediates glia-induced neurodegeneration in Alexander disease. *Nat. Commun.* **6**, 8966 (2015).
- Zamanian, J.L. *et al.* Genomic analysis of reactive astrogliosis. *J. Neurosci.* **32**, 6391-6410 (2012).

5. *If the authors want to claim that increased expression of wild-type GFAP alters mechanotransduction then they need to conduct studies by over-expressing WT GFAP. Again it is not appropriate for the authors here to try and further this notion by disguising the fact that they looked only at mutant-GFAP in this study.*

As suggested, we now show data on expression of wild type GFAP (Supplementary Fig. 4a-d).

6. *In their discussion, the authors selectively cherry pick the literature to cite the only paper that reports any form of mechanistic evidence for a toxic astrocyte phenotype (ref # 49). In doing so, they are being disingenuous. One has to wonder if the present authors even read more than the title of that paper? That paper, in contrast to this one, is written in a scholarly and balanced manner. It refers equally to protective (A2) and toxic (A1) functions of reactive astrocytes and discusses literature providing evidence for both.*

Indeed, evidence from that lab (Barres lab) shows that protective (A2) astrocytes express GFAP levels equal to or greater than toxic (A1) ones (see Zamanian et al paper from the Barres lab). Indeed, there is ample evidence that reactive astrocytes that have clearly been demonstrated to be protective express equal levels of GFAP to those that are implicated as being toxic. The authors should discuss this. In addition, there are multiple studies in which deletion of GFAP results in more, not less, tissue damage after stroke or in mice with Alzheimer's pathology. (see Li L et al. (2008) Protective role of reactive astrocytes in brain ischemia. J Cereb Blood Flow Metab 28:468-481. – and Kraft et al, (2013) Attenuating astrocyte activation accelerates plaque pathogenesis in APP/PS1 mice. FASEB J 27:187-198.) The authors here need to read the literature better and present a more balanced and scholarly introduction and discussion. The last paragraph of the discussion is truly a one-sided over-reach that is not at all justified by the data in this study.

We appreciate the Reviewer's critical discussion of these points and highlighting of relevant references. We have now cited and discussed all these references in our manuscript.

7. Last but not least, the authors should check the literature better on tissue stiffness after CNS injury. A paper recently published in this journal presents results that have bearing on the work presented here. Moeendarbary et al (2017) The soft mechanical signature of glial scars in the central nervous system. Nature communications 8:14787. (I am not a co-author). This study uses atomic force microscopy and convincingly shows that glial scars in vivo were unexpectedly softer (not stiffer) than uninjured tissue. Glial scars are formed largely by astrocytes that express among the highest levels of GFAP encountered after any form of pathology. This observation does not fit with the contention by the present authors that upregulated expression of normal GFAP massively increases tissue stiffness, and suggests that instead it is the AxD-mutant-GFAP that is causing this. The authors here need to discuss this point as well, and not ignore it.

The reference the Reviewer cites is quite interesting. The authors performed a stab injury and then examined an approximately 150 μm area around the site of injury. There was a three-fold drop in elasticity at 9 days following injury and two-fold drop at 22 days following injury. Unfortunately, standard histological sections correlated to regions of tissue stiffness measurements were not provided. However, examination of Fig. 2a,c suggests that measurements provided may reflect the properties of a loose, macrophage-rich infarct cavity as would be expected by the mode and timing of injury. Indeed, a similar study performed by Saxena et al. (2012) demonstrated loss of myelin and cyst formation, both of which may have influenced tissue stiffness measurements. These findings would be consistent with the longstanding neuropathological observation that acute infarcts and organizing and organized infarct cavities are grossly soft, while brains with diffuse longstanding gliosis are grossly firm. Since additional data would be required to confirm these speculations, we feel it is most appropriate to simply comment in the Discussion that brain stiffness following injury may depend on the timing and mode of injury.

Reference:

Saxena, T., Gilbert, J., Stelzner, D. & Hasenwinkel, J. Mechanical characterization of the injured spinal cord after lateral spinal hemisection injury in the rat. *J. Neurotrauma*. **29**, 1747-1757 (2012).

Reviewers' comments:

Reviewer #1 (Remarks to the Author):

The authors have addressed most of the comments of the reviewers. I still think that the upregulation of YAP/Yki and lamin A/C / LamC in AD as well as the link between GFAP, spectraplakin and actin are very interesting. However, despite an impressive number of additional experiments, the authors currently fail to address some of the key points.

The authors still over-interpret their data. For example, in the section called 'Altered cytoskeletal mechanotransduction' all that is shown is a correlation between AD and expression levels of proteins that may or may not be involved in mechanotransduction. There is no direct evidence for an involvement of mechanotransduction in the disease.

I am still not happy with the brain stiffness measurements the authors present. The sample overfill may very well change results, as the dissipation of the stress will be different. I agree though that, given the small mismatch that is now illustrated in the new figure, this effect might be minor. However, the laser-based method applied to the drosophila brain is not strong enough to deform tissue. It is an optical tweezers-based system which at best can apply forces up to a few hundred pN, which was enough in the original study to deform individual cell interfaces (i.e., two neighboring cell membranes) but which is certainly not enough to deform a bulk tissue. This is a well-known limitation of the method that can be found in many original papers and reviews. The movie that is shown is not convincing, it seems to consist only of two frames, and the whole tissue seems to move. The maximum deformation seems to be at the edge of the image, where very likely the laser was not placed. It seems very likely that what is shown is an artifact.

I am also not convinced that embedding brains in collagen leads to their stiffening or to increased cellular forces (I would not know how), unless this is shown in an experiment. Hence, I don't think the statement 'These findings demonstrate that Drosophila brains are indeed responsive to mechanical stimuli' is backed up by the current experimental data.

I also agree with reviewer 3 about the selectivity of citations and their interpretation. In the paper mentioned by that reviewer (Moeendarbary et al (2017)), the authors show a clear correlation between an increase in GFAP levels and tissue softening. This needs to be discussed. The measurements done in the Saxena paper mentioned in the rebuttal were done on tissue replacing spinal cord tissue after it was completely removed, which should thus not be relevant in this context.

A more critical discussion of the results of the current study, which is novel and very interesting, would significantly improve the current manuscript.

Reviewer #2 (Remarks to the Author):

Further studies added to the manuscript strengthen the conclusion that expressing mutant GFAP in fly glia reorganizes actin, which in turn produces an increase in LamC levels via an increase in Hpo/YAP signaling. The authors have found an interesting set of mechanisms and dynamics that link the mutant GFAP to significant cytoskeletal alterations. I still have questions about two of the conclusions, principally because the mechanistic links between GFAP accumulation and cell death and between GFAP accumulation and Hpo/YAP activation are not yet clear and there could be other explanations.

1. The authors define GFAP toxicity in Drosophila primarily by assessing cell death.

a. TUNEL labeling combined with neuronal and glial markers shows death of both cell types in flies expressing mutant GFAP. About 30% of the cell death is neuronal (Suppl Fig 2m), the rest in non-neuronal cells. This tells us that expressing the R79H GFAP in fly glia kills glial cells as well as neurons. The increase in LamC levels in glia contributes significantly to the neuronal loss, since reducing LamC expression in glia rescues many of the neurons. However, reducing LamC levels in glia may also reduce glial death, although that is not explicitly stated.

b. Reducing Hpo expression in the R79H flies increased cell death. Presumably this death includes both neurons and glia, although not stated. The authors then conclude that the GFAP mutant activates Hpo expression and downstream targets, including the increase of LamC expression and protein levels, leading to a higher level of stiffness and more cell death.

c. Thus, increasing LamC in glia contributes to glial cell death. This raises the important question of whether the glial cell death itself contributes significantly to the neuronal death. Could the glial cell death contribute significantly to neuronal death by mechanisms unrelated to changes in mechanical "stiffness" of the glia? Alternatively, is it the surviving glia, somehow altered or distorted or stiffened by increases in LamC, cause neuronal death by mechanisms related to mechanotransduction or even outside that of mechanotransduction effects?

2. The authors provide a model of cytoskeletal changes caused by GFAP that eventually lead to Hpo/YAP signaling. This model is consistent with known interactions between F actin and Hpo activation. However, might there be other pathways through which the accumulation of mutant and/or wt GFAP could activate Hpo/YAP? For example, cellular energy stress activates YAP (Mo JS Nat Cell Biol 2015 17:500). The unfolded protein response, which could be activated by mutant GFAP accumulation, also activates hippo in some cell types (Wu H Nat Comm 2015 6:6239 – these authors also note that YAP is degraded by the proteasome). I am not suggesting that that authors solve these questions in this paper, but they should include a discussion of other cellular alterations promoted by mutant GFAP accumulation that could activate the same pathway.

Reviewer #3 (Remarks to the Author):

In this revised manuscript, the authors have appropriately dealt with most of my concerns. They now present a much more balanced view about the potential functions and dysfunctions of astrogliosis and GFAP upregulation. I think the data presented are of good quality and the paper in its current form represents a valuable addition, in particular as regards the potential roles of mechanotransduction signaling in the CNS responses to injury and disease.

Nevertheless, I do have one more small but important point that requires a change in the text.

The second paragraph of the introduction contains the following sentences two sentences:

"Similarly, overexpression of human wild type GFAP in mice causes robust Rosenthal fiber formation, and models other aspects of Alexander disease as well (12-23). Since levels of GFAP overexpression are comparable between wild type GFAP transgenic mice (13) and those that have been observed in reactive gliosis (24), these findings raise the possibility that increased expression of wild type GFAP may cause cellular toxicity in some human brain disorders."

However, the claim that "levels of GFAP overexpression are comparable between wild type GFAP transgenic mice (13) and those that have been observed in reactive gliosis (24)" is not appropriate. This claim is misleading for various reasons, foremost because the two references cited, 13 and 24, do not at all support it. The two papers are not at all comparable. Reference 13 examines only the effect of over-expression of human GFAP. Reference 24 examines mRNA levels of endogenous mouse Gfap after two naturally occurring insults. Neither paper directly compares Gfap levels induced by genetic

over-expression versus naturally occurring gliosis, and neither paper quantifies actual GFAP protein levels. Reference 13 only presents estimates of human GFAP RNA expression in relative terms described as +, ++ and +++, and says that mice with +++ levels die at a few weeks of age. It does not quantitatively compare the levels of human GFAP levels required to drive Rosenthal fibers, but appears to show only data from +++ expressors. It is not appropriate to claim that massive and lethal levels of human GFAP over-expressed in mice are equivalent to levels of mouse GFAP seen after naturally occurring gliosis.

These authors have never shown that over-expression of mouse GFAP is in any way toxic in mice. In this regard it is noteworthy that Rosenthal fibers are not a hallmark of reactive gliosis induced by any naturally occurring insults in mice, such as stroke, trauma, infection or neurodegenerative disease. I know of no neuropathological studies suggesting that this might be the case. If the authors know of such publications, then they should cite them. If not, then this claim should be deleted and this text altered along the lines of:

“Similarly, pronounced overexpression of human wild type GFAP in mice causes robust Rosenthal fiber formation, and models other aspects of Alexander disease as well (12-23), raising the possibility that prolonged overexpression of wild type human GFAP may cause cellular toxicity in some human brain disorders.”

Being precise in language and claims made in citing literature to build scientific arguments is something that matters. The authors may wish to believe that levels of GFAP induced by natural responses to injury or disease can be toxic, but they have not demonstrated that, and they should not cite literature inappropriately to try and strengthen their case. Perhaps human GFAP is different from mouse GFAP. Then they should explore that more as well.

Response to Reviewers

The Reviewer's comments are reproduced in full (italics) followed by our responses (plain text).

Reviewers' comments:

Reviewer #1 (Remarks to the Author):

The authors have addressed most of the comments of the reviewers. I still think that the upregulation of YAP/Yki and lamin A/C / LamC in AD as well as the link between GFAP, spectraplakins and actin are very interesting. However, despite an impressive number of additional experiments, the authors currently fail to address some of the key points.

The authors still over-interpret their data. For example, in the section called 'Altered cytoskeletal mechanotransduction' all that is shown is a correlation between AD and expression levels of proteins that may or may not be involved in mechanotransduction. There is no direct evidence for an involvement of mechanotransduction in the disease.

We appreciate the Reviewer's positive evaluation of the importance of the signaling pathway changes documented in our work, and also understand the concerns regarding our interpretations of these data. In response, we have carefully revised our work to focus on the strengths of our study and avoid overinterpretations. In particular, we have changed the title of the section noted to "Altered cytoskeletal mechanotransduction signaling" to summarize our data more accurately.

I am still not happy with the brain stiffness measurements the authors present. The sample overfill may very well change results, as the dissipation of the stress will be different. I agree though that, given the small mismatch that is now illustrated in the new figure, this effect might be minor. However, the laser-based method applied to the drosophila brain is not strong enough to deform tissue. It is an optical tweezers-based system which at best can apply forces up to a few hundred pN, which was enough in the original study to deform individual cell interfaces (i.e., two neighboring cell membranes) but which is certainly not enough to deform a bulk tissue. This is a well-known limitation of the method that can be found in many original papers and reviews. The movie that is shown is not convincing, it seems to consist only of two frames, and the whole tissue seems to move. The maximum deformation seems to be at the edge of the image, where very likely the laser was not placed. It seems very likely that what is shown is an artifact.

We apologize for not describing our laser-based method more accurately and completely. The typical force used in such experiments (up to 300 mW in Bambardekar et al., 2015) is almost an order of magnitude lower than that used in our studies (2500 mW). Further, we agree that we are causing and measuring local tissue deformation in the area probed by the laser (about 48 μm x 37 μm), rather than moving the entire brain. We have modified our schematic to clarify these points (Fig. 5c). In all experiments (both control and experimental groups), we used the same applied force induced by the laser tweezer. For each measurement, we recorded time-lapse imaging with the setting of 33 frames per second for approximately 3 to 4 seconds. Due to the large size of the resultant video, we showed only two frames before and after tissue deformation as an example in the original supplementary figure, which we now realize was not a clear and adequate presentation of the data. We have now modified our Methods and Supplementary Data to present a substack of 10 raw frames covering the deformation process

(Movie 2). To better illustrate the deformation magnitude, we also included a PIV-processed image which shows the direction and magnitude of a deformation field (Supplementary Fig. 6b). We consider the movement of cells in our experiment to be deformation and not artifact (such as drift movement) because different locations in the videos have different movement directions and magnitudes (Supplementary Fig. 6b). The majority of the videos have a center (usually near the middle of each frame) where the deforming magnitude is minimal locally and increases with the distance from the center, consistent with the radially symmetric nature of the laser trap in such experimental settings (Jahnel et al., 2011; Yehoshua et al., 2015). Finally, to rigorously calculate the magnitude of deformation and remove possible drift displacement, we reanalyzed our data using the method below. The rationale of this method is that each displacement field is a superposition of deformation and drift movement (though small in our videos). By subtracting the drift movement from the displacement field, we can extract the deformation of the tissue from displacement field (Reddy, J. 2013). The algorithm is described as below:

There are $m * n$ points on the PIV results, each point is labelled by using its coordinate at the video, which is (i, j) . (u_i, v_j) are the displacement at coordinate (i, j)

the mean drift displacement in x direction is calculated as $u_{shift} = \frac{\sum_{i=1}^m \sum_{j=1}^n u_{i,j}}{m * n}$

the mean drift displacement in y direction is calculated as $v_{shift} = \frac{\sum_{i=1}^m \sum_{j=1}^n v_{i,j}}{m * n}$

The mean magnitude of deformation is calculated as $d = \frac{\sum_{i=1}^m \sum_{j=1}^n \sqrt{(u_{i,j} - u_{shift})^2 + (v_{i,j} - v_{shift})^2}}{m * n}$

The unit of the magnitude of deformation given in PIV analysis is pixel. One pixel represents 0.04857 μm .

We would further like to clarify that we used the laser-based approach only after considerable effort testing alternative methodologies. The small size of the adult *Drosophila* brain (typical size: 590 $\mu\text{m} \times 340 \mu\text{m} \times 120 \mu\text{m}$, Peng et al., 2011) and irregular 3-dimensional topography made it challenging for us to identify and utilize an appropriate technique to measure tissue stiffness. We considered standard bulk rheology. The smallest rheometer geometry available to us is a 5 mm plate, which we used to measure tissue stiffness in the mouse model of Alexander disease, but is too large for *Drosophila* brain. Further, the small size of *Drosophila* brain makes it challenging to obtain a measurable signal with a traditional rheometer. We then attempted to measure stiffness by deforming the fly brain locally with a trapped bead in an optical tweezer system, but could not use the optical tweezer strategy because the surface of the brain is not flat and beads often became trapped within the tissue. Thus, we ultimately employed the strategies presented in the paper. While perhaps not standard, we believe that we have considered potential pitfalls and alternative interpretations of these approaches in a rigorous manner and hope we have now explained our experimental systems and rationale in a clearer and more comprehensive fashion.

References:

Bambardekar, K., Clément, R., Blanc, O., Chardès, C. & Lenne, P.F. Direct laser manipulation reveals the mechanics of cell contacts in vivo. *Proc. Natl. Acad. Sci. U. S. A.* **112**, 1416-1421 (2015).

Jahnel, M., Behrndt, M., Jannasch, A., Schäffer, E. & Grill, S.W. Measuring the complete force field of an optical trap. *Opt Lett.* **36**, 1260-1262 (2011).

Peng et al. BrainAligner: 3D Registration Atlases of *Drosophila* Brains. *Nat. Methods.* **8**, 493–500 (2011).

Reddy, J. *An Introduction to Continuum Mechanics.* (Cambridge Univ. Press, Cambridge, 2013) 88-97.

Yehoshua, S. Pollari, R. & Milstein, J.N. Axial Optical Traps: A New Direction for Optical Tweezers. *Biophys J.* **108**, 2759-2766 (2015).

I am also not convinced that embedding brains in collagen leads to their stiffening or to increased cellular forces (I would not know how), unless this is shown in an experiment. Hence, I don't think the statement 'These findings demonstrate that Drosophila brains are indeed responsive to mechanical stimuli' is backed up by the current experimental data.

Again, given the challenges of applying conventional methodologies to the small fly brain, we decided to use a strategy novel to the fly brain, but well-established in cell and organ culture as a method of assessing the response to mechanical force. Since culture on substrates of different stiffness is a widely accepted method for probing the response of cells to different amounts of mechanical stress (Discher et al., 2005; Baker et al., 2009; Dupont et al., 2011), we assume that the Reviewer is questioning the application of the approach to a more complex tissue. In response, we now note that culture in matrices of different stiffness has also been used in multiple organoid culture systems to study the role of tissue stiffness in biological responses of various sorts. For example, in mammary epithelium organoids the stiffness of the extracellular matrix regulates the induction of malignant phenotypes and tumor progression, which is mediated by integrin focal adhesion signaling (Levental et al., 2009; Chaudhuri et al., 2014). Such organoids vary in size, but may be up to hundreds of microns. Since we have been able to show, using transmission electron microscopy, that the collagen-rich outer neural lamella of the fly brain interacts directly with collagen fibrils in the matrix (Supplementary Fig. 6c,d, yellow arrowheads), we believe that the small cultured fly brain is plausibly subjected to varying physical force in collagen matrix of different stiffness. However, we agree that other interpretations of our data are possible, including force-independent, collagen concentration dependent signaling, and we have thus been more circumspect in the interpretation of our findings.

References:

Baker, E.L., Bonnecaze, R.T. & Zaman, M.H. Extracellular matrix stiffness and architecture govern intracellular rheology in cancer. *Biophys. J.* **97**, 1013-1021 (2009).

Discher, D.E., Janmey, P. & Wang, Y.L. Tissue cells feel and respond to the stiffness of their substrate. *Science.* **310**, 1139-1143 (2005).

Dupont, S. *et al.* Role of YAP/TAZ in mechanotransduction. *Nature.* **474**, 179-183 (2011).
Chaudhuri, O., et al. Extracellular matrix stiffness and composition jointly regulate the induction of malignant phenotypes in mammary epithelium. *Nat. Mater.* **13**, 970-978 (2014).

Levental, K.R. et al. Matrix crosslinking forces tumor progression by enhancing integrin signaling. *Cell*. **139**, 891-906 (2009).

I also agree with reviewer 3 about the selectivity of citations and their interpretation. In the paper mentioned by that reviewer (Moeendarbary et al (2017)), the authors show a clear correlation between an increase in GFAP levels and tissue softening. This needs to be discussed. The measurements done in the Saxena paper mentioned in the rebuttal were done on tissue replacing spinal cord tissue after it was completely removed, which should thus not be relevant in this context.

We appreciate the Reviewers' providing their feedback that our manuscript appeared unbalanced in our citations and interpretations. We have done our best to revise the manuscript to address these concerns within the constraints on the number of references allowed by the journal. Specifically, we have referenced and discussed the work of Moeendarbary and colleagues (2017). Perhaps our discussion of the Saxena paper was unnecessarily distraction. Our point was that tissue stiffness after injury, particularly a destructive injury, will reflect not only the expression level of GFAP within astrocytes, but also the degree and nature of tissue damage, macrophage infiltration with accompanying release of proteases, debris clearance and ultimately cavity formation. We note that the nature and time course of the penetrating tissue injury in the Moeendarbary study is consistent with accompanying tissue softening due to factors extrinsic to, and possibly independent of, astrocytic GFAP upregulation.

A more critical discussion of the results of the current study, which is novel and very interesting, would significantly improve the current manuscript.

Again, we appreciate the positive evaluation of our manuscript and suggestions for improvement and have done our best to provide a more balanced and critical discussion.

Reviewer #2 (Remarks to the Author):

Further studies added to the manuscript strengthen the conclusion that expressing mutant GFAP in fly glia reorganizes actin, which in turn produces an increase in LamC levels via an increase in Hpo/YAP signaling. The authors have found an interesting set of mechanisms and dynamics that link the mutant GFAP to significant cytoskeletal alterations. I still have questions about two of the conclusions, principally because the mechanistic links between GFAP accumulation and cell death and between GFAP accumulation and Hpo/YAP activation are not yet clear and there could be other explanations.

*1. The authors define GFAP toxicity in Drosophila primarily by assessing cell death.
a. TUNEL labeling combined with neuronal and glial markers shows death of both cell types in flies expressing mutant GFAP. About 30% of the cell death is neuronal (Suppl Fig 2m), the rest in non-neuronal cells. This tells us that expressing the R79H GFAP in fly glia kills glial cells as well as neurons. The increase in LamC levels in glia contributes significantly to the neuronal loss, since reducing LamC expression in glia rescues many of the neurons. However, reducing LamC levels in glia may also reduce glial death, although that is not explicitly stated.*

The Reviewer is correct that modulation of LamC expression in glial cells affects both glial and neuronal cell death. When we increased LamC levels in glia, we observed both glial and neuronal cell death; about 30% of apoptotic cells are neurons and the remainder glia

(Supplementary Fig. 2k-m). When we reduced LamC expression, we also rescued both glial and neuronal cell death (Fig. 2g,i). We have modified the text and figures to make this point more clearly.

b. Reducing Hpo expression in the R79H flies increased cell death. Presumably this death includes both neurons and glia, although not stated. The authors then conclude that the GFAP mutant activates Hpo expression and downstream targets, including the increase of LamC expression and protein levels, leading to a higher level of stiffness and more cell death.

We agree with the Reviewer and have edited the manuscript for increased clarity on this point.

c. Thus, increasing LamC in glia contributes to glial cell death. This raises the important question of whether the glial cell death itself contributes significantly to the neuronal death. Could the glial cell death contribute significantly to neuronal death by mechanisms unrelated to changes in mechanical “stiffness” of the glia? Alternatively, is it the surviving glia, somehow altered or distorted or stiffened by increases in LamC, cause neuronal death by mechanisms related to mechanotransduction or even outside that of mechanotransduction effects?

We agree with the Reviewer, who raises a number of interesting points. In this manuscript we have focused primarily on glial-cell intrinsic events that control nervous system toxicity in our glial GFAP transgenic animals, including secondary non-cell autonomous death of neurons. We believe that our genetic evidence strongly supports a causative role for lamin and the hippo pathway in glia controlling toxicity in our system. The robust genetic modification provided by lamin and hippo pathway modulations argues for a key role of the pathway, but we cannot exclude a more modest contribution from another pathway(s). The exact mechanisms leading to non-cell autonomous neuronal death will require additional investigation. We have previously demonstrated that aberrant glial-neuronal signaling plays a role in Alexander disease (Wang et al., 2015). However, the full complement of signaling mechanisms controlling the interaction between the two cell types in Alexander disease likely remains to be defined. Death of glia may also contribute to neuronal death. A simple experiment to block death of glia in our Alexander disease model resulted in interesting, but complex, effects on neuronal and glial viability. We are currently in the process of working to understand these results mechanistically, but significant additional effort will be needed before we can present a full and complete description of these findings.

2. The authors provide a model of cytoskeletal changes caused by GFAP that eventually lead to Hpo/YAP signaling. This model is consistent with known interactions between F actin and Hpo activation. However, might there be other pathways through which the accumulation of mutant and/or wt GFAP could activate Hpo/YAP? For example, cellular energy stress activates YAP (Mo JS Nat Cell Biol 2015 17:500). The unfolded protein response, which could be activated by mutant GFAP accumulation, also activates hippo in some cell types (Wu H Nat Comm 2015 6:6239 – these authors also note that YAP is degraded by the proteasome). I am not suggesting that that authors solve these questions in this paper, but they should include a discussion of other cellular alterations promoted by mutant GFAP accumulation that could activate the same pathway.

We thank the reviewer for raising these helpful points and have expanded our discussion accordingly.

Reviewer #3 (Remarks to the Author):

In this revised manuscript, the authors have appropriately dealt with most of my concerns. They now present a much more balanced view about the potential functions and dysfunctions of astrogliosis and GFAP upregulation. I think the data presented are of good quality and the paper in its current form represents a valuable addition, in particular as regards the potential roles of mechanotransduction signaling in the CNS responses to injury and disease.

Nevertheless, I do have one more small but important point that requires a change in the text.

The second paragraph of the introduction contains the following sentences two sentences: “Similarly, overexpression of human wild type GFAP in mice causes robust Rosenthal fiber formation, and models other aspects of Alexander disease as well (12-23). Since levels of GFAP overexpression are comparable between wild type GFAP transgenic mice (13) and those that have been observed in reactive gliosis (24), these findings raise the possibility that increased expression of wild type GFAP may cause cellular toxicity in some human brain disorders.”

However, the claim that “levels of GFAP overexpression are comparable between wild type GFAP transgenic mice (13) and those that have been observed in reactive gliosis (24)” is not appropriate. This claim is misleading for various reasons, foremost because the two references cited, 13 and 24, do not at all support it. The two papers are not at all comparable. Reference 13 examines only the effect of over-expression of human GFAP. Reference 24 examines mRNA levels of endogenous mouse Gfap after two naturally occurring insults. Neither paper directly compares Gfap levels induced by genetic over-expression versus naturally occurring gliosis, and neither paper quantifies actual GFAP protein levels. Reference 13 only presents estimates of human GFAP RNA expression in relative terms described as +, ++ and +++, and says that mice with +++ levels die at a few weeks of age. It does not quantitatively compare the levels of human GFAP levels required to drive Rosenthal fibers, but appears to show only data from +++ expressors. It is not appropriate to claim that massive and lethal levels of human GFAP over-expressed in mice are equivalent to levels of mouse GFAP seen after naturally occurring gliosis.

These authors have never shown that over-expression of mouse GFAP is in any way toxic in mice. In this regard it is noteworthy that Rosenthal fibers are not a hallmark of reactive gliosis induced by any naturally occurring insults in mice, such as stroke, trauma, infection or neurodegenerative disease. I know of no neuropathological studies suggesting that this might be the case. If the authors know of such publications, then they should cite them. If not, then this claim should be deleted and this text altered along the lines of:

“Similarly, pronounced overexpression of human wild type GFAP in mice causes robust Rosenthal fiber formation, and models other aspects of Alexander disease as well (12-23), raising the possibility that prolonged overexpression of wild type human GFAP may cause cellular toxicity in some human brain disorders.”

Being precise in language and claims made in citing literature to build scientific arguments is something that matters. The authors may wish to believe that levels of GFAP induced by natural responses to injury or disease can be toxic, but they have not demonstrated that, and they should not cite literature inappropriately to try and strengthen their case. Perhaps human GFAP is different from mouse GFAP. Then they should explore that more as well.

We have modified the text in accordance with the Reviewer’s suggestion.

Reviewers' comments:

Reviewer #1 (Remarks to the Author):

The authors have now addressed all reviewer concerns, and the manuscript has significantly improved. I have, however, to re-raise one major and a few minor points that still need to be addressed.

Major point:

As the authors have now provided experimental details of the laser-based deformation assay, I am sure that this approach is not suited to measure tissue stiffness. Using 2.5W of a 1064nm laser will undoubtedly lead to a massive heating of the sample (up to 40 K/W!) (e.g., Català et al 2017, and many other papers). There is a reason why optical traps are usually limited to few hundred milliwatts when used with biological samples, and why only cell membranes are usually deformed. Also, unless the tissue has a lower refractive index than its environment (which is highly unlikely), it should be 'pulled' towards higher laser powers, i.e., towards the center of the laser beam. However, in the movie tissue rather seems to be pushed away (the arrows in fig. S6b aren't really visible). This is much more likely the consequence of the expansion of the heated water in the sample. Hence, these measurements are misleading and should be omitted. However, I don't think this leads to a change in the story. The authors show data about brain mechanics from the mouse model, which I feel are sufficient.

Minor points:

1. The authors have addressed some of the major overinterpretations and have toned their statements down. However, a few statements are left that are not backed up by data and should thus be toned down too.
 - a. 'As an additional method of altering mechanical stress, we performed ex vivo 3-dimensional (3D) collagen matrix culture of fly brains.' As far as I can tell, the authors haven't performed a single experiment to alter mechanical stresses (stress = force per unit area). If external forces have been applied to the matrices, this should be mentioned. The authors have done one valid measurement of brain tissue stiffness, and now they are putting whole brains into a collagen matrix hoping that brain cells respond to its mechanical properties. This is also how it should be stated.
 - b. 'These findings suggest that Drosophila brains are responsive to mechanical stimuli and that LamC and Yki respond directly to culture matrix stiffness.' Again, I am not convinced about this point. I do realize that this approach may work in some organoid cultures, which however, are very different in nature than intact organs. The attachment of collagen fibrils to neural lamella is very indirect; to show that there are mechanical interactions between brain and matrix a deformation of the matrix by the brain would need to be shown. However, this is certainly beyond the scope of the current study. What needs to be done is a more careful interpretation of the data.
 - c. In the discussion, the authors state that they 'provide strong evidence for altered mechanotransduction in the primary astrogliaopathy Alexander disease'. As mentioned before, the authors do not present any direct evidence of altered mechanotransduction. In order to show that, they would need to perturb mechanical signals, which was not done. What they show is an upregulation of proteins that are potentially involved in mechanotransduction (but also other) pathways, and a change in brain tissue stiffness, which might or might not be related, and which might be a cause or a consequence of the disease progression. A more accurate description of their work, which is novel and exciting, would, for example, be '... to obtain first insights into a potentially new mechanism of the primary astrogliaopathy Alexander disease, involving changes in brain tissue stiffness and expression levels of proteins that are found in mechanotransduction cascades.'
 - d. Similarly, the authors state 'By providing genetic and biophysical evidence linking altered mechanotransduction signaling with behavioral and cellular toxicity in Alexander disease'. Again, there

is no direct evidence for altered mechanotransduction. Needs to be rephrased.

2. The authors state 'We measured the tissue elasticity of 750 μm -thick brain sections from the hippocampus'. They also stated in their rebuttal that the tissue fitted a 5mm geometry. I assume what they measured is brain tissue containing hippocampal tissue? Murine hippocampal tissue wouldn't fit the geometry (leading to sample underfill as mentioned before). This should clearly be stated.

3. In the experiments with human IPS cells, it needs to be clearly stated that those are cultures on conventional tissue culture plastics to avoid confusion.

4. 'We propose that the concerted effects of altered mechanotransduction signaling then combine to promote increased brain stiffness (Fig. 5a-d), which in turn sensitizes brains to additional mechanical stress (Fig. 5e-j).' An increase in brain stiffness should lead to smaller strains in response to an applied stress. Hence, it is not clear to me how increased brain stiffness should lead to a sensitization to mechanical stress. The authors might want to reconsider this hypothesis or rephrase it.

Reviewer #3 (Remarks to the Author):

In this additionally revised manuscript, the authors have appropriately dealt with my concerns. I continue to think the paper in its current form represents a valuable addition.

Response to Reviewers

The Reviewer's comments are reproduced in full (*italics*) followed by our responses (*plain text*).

Reviewers' comments:

Reviewer #1 (Remarks to the Author):

The authors have now addressed all reviewer concerns, and the manuscript has significantly improved. I have, however, to re-raise one major and a few minor points that still need to be addressed.

Major point:

As the authors have now provided experimental details of the laser-based deformation assay, I am sure that this approach is not suited to measure tissue stiffness. Using 2.5W of a 1064nm laser will undoubtedly lead to a massive heating of the sample (up to 40 K/W!) (e.g., Català et al 2017, and many other papers). There is a reason why optical traps are usually limited to few hundred milliwatts when used with biological samples, and why only cell membranes are usually deformed. Also, unless the tissue has a lower refractive index than its environment (which is highly unlikely), it should be 'pulled' towards higher laser powers, i.e., towards the center of the laser beam. However, in the movie tissue rather seems to be pushed away (the arrows in fig. S6b aren't really visible). This is much more likely the consequence of the expansion of the heated water in the sample. Hence, these measurements are misleading and should be omitted. However, I don't think this leads to a change in the story. The authors show data about brain mechanics from the mouse model, which I feel are sufficient.

Based on the comments of the Reviewer and Editor, we have removed the laser trap experiments from the manuscript.

Minor points:

1. The authors have addressed some of the major overinterpretations and have toned their statements down. However, a few statements are left that are not backed up by data and should thus be toned down too.

a. 'As an additional method of altering mechanical stress, we performed ex vivo 3-dimensional (3D) collagen matrix culture of fly brains.' As far as I can tell, the authors haven't performed a single experiment to alter mechanical stresses (stress = force per unit area). If external forces have been applied to the matrices, this should be mentioned. The authors have done one valid measurement of brain tissue stiffness, and now they are putting whole brains into a collagen matrix hoping that brain cells respond to its mechanical properties. This is also how it should be stated.

We have modified the text to reflect the Reviewer's comments.

b. 'These findings suggest that Drosophila brains are responsive to mechanical stimuli and that LamC and Yki respond directly to culture matrix stiffness.' Again, I am not convinced about this point. I do realize that this approach may work in some organoid cultures, which however, are very different in nature than intact organs. The attachment of collagen fibrils to neural lamella is very indirect; to show that there are mechanical interactions between brain and matrix a deformation of the matrix by the brain would need to be shown. However, this is certainly beyond the scope of the current study. What needs to be done is a more careful interpretation of the data.

We have revised our statements to reflect the Reviewer's points.

c. In the discussion, the authors state that they 'provide strong evidence for altered mechanotransduction in the primary astrogliaopathy Alexander disease'. As mentioned before, the authors do not present any direct evidence of altered mechanotransduction. In order to show that, they would need to perturb mechanical signals, which was not done. What they show is an upregulation of proteins that are potentially involved in mechanotransduction (but also other) pathways, and a change in brain tissue stiffness, which might or might not be related, and which might be a cause or a consequence of the disease progression. A more accurate description of their work, which is novel and exciting, would, for example, be '... to obtain first insights into a potentially new mechanism of the primary astrogliaopathy Alexander disease, involving changes in brain tissue stiffness and expression levels of proteins that are found in mechanotransduction cascades.'

We have modified the text as suggested by the Reviewer.

d. Similarly, the authors state 'By providing genetic and biophysical evidence linking altered mechanotransduction signaling with behavioral and cellular toxicity in Alexander disease'. Again, there is no direct evidence for altered mechanotransduction. Needs to be rephrased.

We have rephrased the statement for accuracy.

2. The authors state 'We measured the tissue elasticity of 750 μm -thick brain sections from the hippocampus'. They also stated in their rebuttal that the tissue fitted a 5mm geometry. I assume what they measured is brain tissue containing hippocampal tissue? Murine hippocampal tissue wouldn't fit the geometry (leading to sample underfill as mentioned before). This should clearly be stated.

We have made the clarification suggested in the text.

3. In the experiments with human IPS cells, it needs to be clearly stated that those are cultures on conventional tissue culture plastics to avoid confusion.

We have provided additional details regarding the iPSC culture methods, and have specifically indicated that the cells were grown on glass cover slips coated with matrigel prior to the staining experiments described in both the Methods and Results sections.

4. 'We propose that the concerted effects of altered mechanotransduction signaling then combine to promote increased brain stiffness (Fig. 5a-d), which in turn sensitizes brains to additional mechanical stress (Fig. 5e-j).' An increase in brain stiffness should lead to smaller strains in response to an applied stress. Hence, it is not clear to me how increased brain stiffness should lead to a sensitization to mechanical stress. The authors might want to reconsider this hypothesis or rephrase it.

Thank you for raising this point. We have revised our text accordingly.

Reviewer #3 (Remarks to the Author):

In this additionally revised manuscript, the authors have appropriately dealt with my concerns. I

continue to think the paper in its current form represents a valuable addition.

REVIEWERS' COMMENTS:

Reviewer #1 (Remarks to the Author):

The authors have addressed all reviewer comments. The manuscript will be of large interest to a wide readership.